# NovelCraft: A Dataset for Novelty Detection and Discovery in Open Worlds

**Patrick Feeney**[1,*]                                             *patrick.feeney@tufts.edu*
**Sarah Schneider**[1,2,*]
**Panagiotis Lymperopoulos**[1]
**Li-Ping Liu**[1]
**Matthias Scheutz**[1]
**Michael C. Hughes**[1]                                          *michael.hughes@tufts.edu*
[1] *Dept. of Computer Science, Tufts University*
[2] *Center for Vision, Automation and Control, Austrian Institute of Technology*
[*] *Authors PF and SS both provided lead author contributions*

**Reviewed on OpenReview:** *https://openreview.net/forum?id=4eL6z9ziw7*

## Abstract

In order for artificial agents to successfully perform tasks in changing environments, they must be able to both detect and adapt to novelty. However, visual novelty detection research often only evaluates on *repurposed* datasets such as CIFAR-10 originally intended for object classification, where images focus on one distinct, well-centered object. New benchmarks are needed to represent the challenges of navigating the complex scenes of an open world. Our new NovelCraft[1] dataset contains *multimodal* episodic data of the images and symbolic world-states seen by an agent completing a pogo stick assembly task within a modified Minecraft environment. In some episodes, we insert novel objects of varying size within the complex 3D scene that may impact gameplay. Our visual novelty detection benchmark finds that methods that rank best on popular area-under-the-curve metrics may be outperformed by simpler alternatives when controlling false positives matters most. Further multimodal novelty detection experiments suggest that methods that fuse both visual and symbolic information can improve time until detection as well as overall discrimination. Finally, our evaluation of recent generalized category discovery methods suggests that adapting to new imbalanced categories in complex scenes remains an exciting open problem.

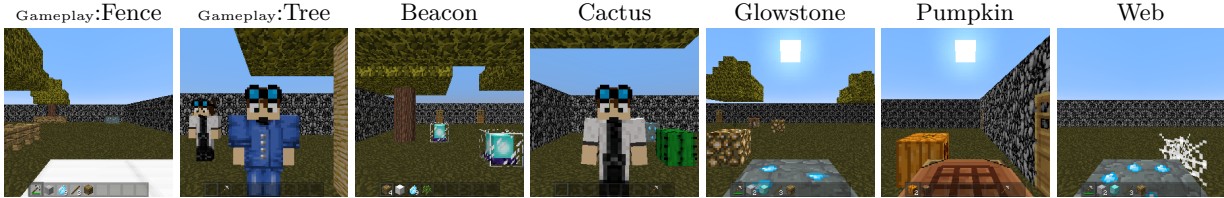

Figure 1: Example images for a subset of novelties in NovelCraft. See Appendix Fig. A.1 for images of all 53 novelties.

## 1 Introduction

Recent progress in computer vision (Krizhevsky et al., 2017) and vision-informed reinforcement learning (Mnih et al., 2015; Silver et al., 2018) is exciting but focused on tasks like classification or video game playing where the agent's goals are narrowly defined in a world that does not change. Many real applications require agents that can navigate *open worlds* that evolve over time (Boult et al., 2019). Designing effective agents for these domains requires the ability to detect, characterize, and ultimately adapt to *novel* stimuli.

---

[1]Open-access dataset [CC-BY License] and code [MIT License]: https://NovelCraft.cs.tufts.edu/

To define novelty, we take the point of view of an agent with some prior experience in a given environment. We define a stimuli as *novel* to the agent if it considerably differs from the agent's model (Ruff et al., 2021). Some authors refer to similar problems as *anomaly detection* or *out-of-distribution detection*. Novelties are not simply outliers. Detecting a novelty suggests that revising the agent's model is necessary, whereas an outlier is a rare event expected (however infrequently) under the existing model.

Work on novelty detection has spanned decades (Markou & Singh, 2003; Pimentel et al., 2014; Ruff et al., 2021), with a wealth of available paradigms spanning "one-class" boundary learning (Schölkopf et al., 2000), reconstruction error (Abati et al., 2019), probabilistic density modeling (Nalisnick et al., 2018), and more. For vision-based novelty detection, recent progress has been dominated by modifications of deep image classifiers (Lakshminarayanan et al., 2017; Liang et al., 2018; Cheng & Vasconcelos, 2021).

Despite this wealth of methods, *evaluation practices* have seen less attention. Recent visual novelty detection systems are typically evaluated by repurposing object-focused datasets such as Caltech-256 (Griffin et al., 2007) or CIFAR-10 (Krizhevsky, 2009) that were originally intended for supervised classification. A typical evaluation paradigm trains on a "normal" subset of available classes, then tries to detect "novel" images from other classes unseen during training. We contend that when using object-focused datasets, this paradigm is too *optimistic*, representing best-case scenarios where the object is relatively large and well-framed within the image. While some efforts have turned toward *fine-grained* classification datasets like distinguishing novel breeds of dogs (Cheng & Vasconcelos, 2021), these images remain object-focused. To develop agents ready to navigate open worlds, we argue that a *scene-focused* benchmark is needed.

We have developed and released a comprehensive scene-focused dataset that we call NovelCraft, for benchmarking systems that can detect and adapt to novelty in open worlds. Our target domain is a modified

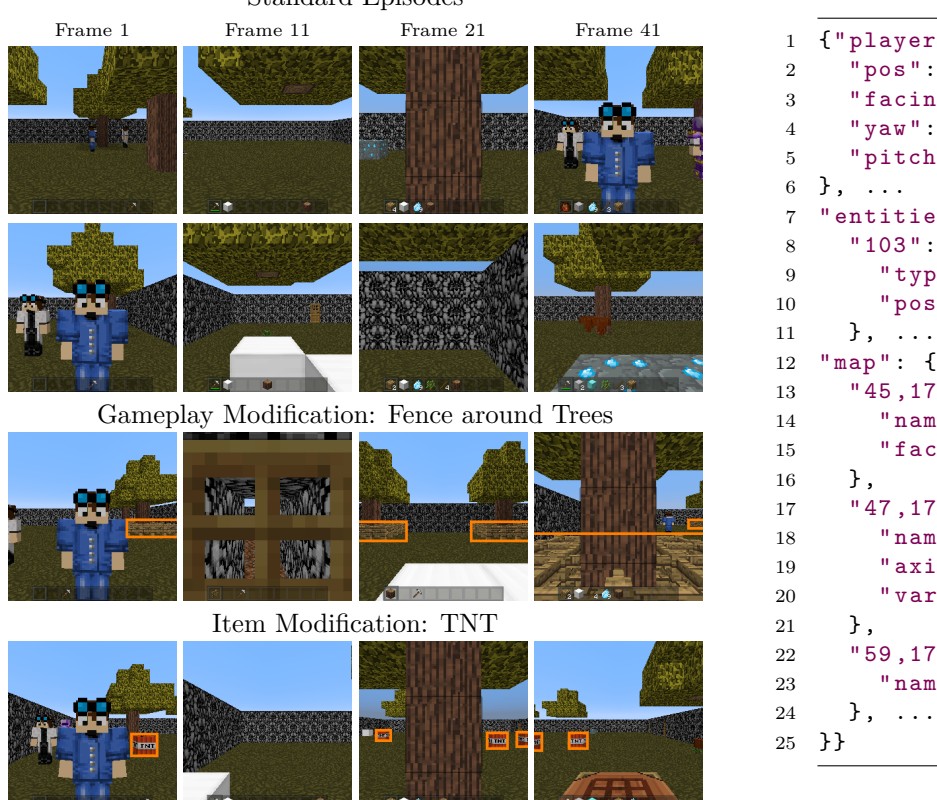

Figure 2: Example contents of our new multi-modal NovelCraft dataset. **Left:** Images from two normal episodes (rows 1-2) and two novel episodes (rows 3-4). Detection is challenging as only a few images in novel episodes actually contain novel objects, here outlined in orange when visible. **Right:** Example symbolic information available at each frame. At each frame, we record a complete JSON representation of the player state and all objects and artificial agents in the world ($x,z,y$ position in 3D, orientation, etc.). This includes objects outside the player's view recorded in images.

version of Minecraft that provides a complex, colorful 3D world with a pixelated video game aesthetic. Within this world, the agent's goal is to use a carefully selected sequence of actions to turn available resources into a pogo stick. We have recorded the observations of an artificial agent running through hundreds of *episodes* of this pogo stick construction task, each one producing a sequence of images as well as non-visual symbolic information about the world's state (including all objects, not just those visible from agent's current viewpoint). In some of these episodes, the game has been deliberately modified so that *novel* types of objects (e.g. TNT explosives, fences, or jukeboxes) are inserted into the world and potentially visible to the agent. Some of these novel objects can impact gameplay (e.g. trees with distinct bark color may yield more resources). Representative images from two "normal" episodes and two "novel" episodes are shown in Fig. 2, visualizing how novel objects vary in placement and size in complex scenes.

Using this data, we define three benchmark tasks. First, we have two flavors of *novelty detection*: (1) using vision (Sec. 4), and (2) using non-visual symbolic or multimodal information (Sec. 5). Next, we consider (3) *characterization* of novelty via a category discovery task (Sec. 6) where the system must find new objects within complex scenes and organize them within an ever-growing system of classes. We benchmark how state-of-the-art methods perform at each task.

***The contributions of this study*** are:

1. **A scene-focused dataset purpose-built for visual novelty detection.** Existing evaluations are too object-focused, artificially creating "novelty" by repurposing images designed for object classification. In Sec. 4, we benchmark visual detectors on our new data of complex scenes containing many objects, with novelties often found in the periphery or occluded by other objects when the agent first encounters them.
2. **Inclusion of two useful modalities: images and symbolic world-state.** In Sec. 5, we show how our data supports visual as well as non-visual symbolic processing to detect novelties. Our dataset can be used to prototype *multimodal* methods, which are thus far under-explored in the literature.
3. **Examination of metrics informed by task-specific costs.** While we do report widely used area-under-curve scores to assess detectors, we contend that reporting *only* these metrics ignores the need to pick a threshold when the system must produce binary decisions. Threshold selection must be guided by the task-specific costs of false positives versus false negatives in any deployment.
4. **Going beyond detection, we assess category discovery.** The ability to recognize a previously seen novelty is necessary for an agent to improve with more experience. In Sec. 6, we show how our dataset can be used to inspire improvements in generalized category discovery and share code and pretrained model for generalized category discovery.

Ultimately, we hope our work is a step toward a long-term goal of agents that fuse vision with other sensors to adapt to open worlds. Our primary motivation is not one specific real application (like self-driving cars) but rather a general style of problem-solving that we think will become increasingly common and useful: the integration of vision systems for complex scenes within the overall cognitive architectures of a goal-oriented agent to successfully adapt to novelty in 3D open worlds over time.

**Rationale for focus on a virtual world.** We specifically selected this Minecraft-like world because it has several key attributes that are helpful toward our long-term goal. First, it requires long-range planning over time toward a clear goal (pogo stick assembly), not just immediate stimuli-driven responses (e.g. avoid an obstacle in the road). Second, gameplay in this world allows interaction with other agents (both beneficial and adversarial). Third, gameplay requires reasoning about a combinatorial action space as well as 3D movement (vertical or horizontal). Our current benchmark and this research field in general are not at the finish line, only a few steps into a longer journey. We posit that our dataset and evaluation practices are a useful step forward, and the virtual platform we build upon can grow towards this goal.

## 2    Related Work on Datasets for Novelties and Anomalies

**Object-focused datasets.** Datasets showcasing distinct object types, such as CIFAR-10, Caltech-256, and ImageNet (Deng et al., 2009), originally intended for supervised classification, are often repurposed to assess novelty detection under a *k-classes-out* paradigm (Ruff et al., 2021). This approach continues to dominate vision research related to novelty detection: The Open World Vision workshop at CVPR 2021 used repurposed images from ImageNet in their open-set recognition challenge (Kumar et al., 2021). Some

| Dataset | Agent pursuing goal? | Images over time? | Multimodal? | Novel classes |
|---|---|---|---|---|
| **NovelCraft** | YES | YES | YES | 53 |
| COCO Lin et al. (2014) | No | No | YES | 91 |
| UBnormal Acsintoae et al. (2022) | No | YES | No | 5 |
| RoadAnomaly Chan et al. (2021) | No | No | No | 26 |

Table 1: Summary of visual datasets focused on complex scenes, not individual objects. (Expanded in Tab. D.1)

recent efforts use synthetic perturbations of ImageNet images to assess robustness (Hendrycks & Dietterich, 2019) and separate novelties based on the level of semantic similarity to the training classes (Vaze et al., 2021). Many recent novelty detection methods, such as work by Cheng & Vasconcelos (2021), have yet to be evaluated on *scenes*, where novelties may not be large, centered or fully visible in the images.

**Scene-focused datasets.** Some recent work has developed datasets that could be used to assess novelty or anomalies in complex visual scenes. The Common Objects in Context (COCO-2014) dataset (Lin et al., 2014) is a widely-used benchmark for scene understanding containing complex images gathered from Flickr, but was not intended to assess novelty detection or discovery. The UBnormal dataset (Acsintoae et al., 2022) contains fixed perspective videos of complex virtual scenes designed to contain anomalous events. Several efforts look at anomalies in scenes gathered from autonomous driving, such as Fishyscapes (Blum et al., 2021) or Road Anomaly (Lis et al., 2019; Chan et al., 2021).

Table 1 compares these datasets to NovelCraft. An expanded table assessing even more datasets is available in Appendix Table D.1. As seen in these tables, NovelCraft is the only dataset containing egocentric images from a goal-oriented agent exploring a 3D world over time. NovelCraft supports benchmark tasks for both detection and category discovery with 50+ classes, while other datasets offer only a limited number of object classes to be discovered ($\leq$10 in UBnormal and Fishyscapes, <30 for RoadAnomaly). COCO, which does provide more classes, is not built for novelty detection, does not have images over time from an agent pursuing a goal, and does not have defined dataset splits for novelty detection or category discovery. Finally, NovelCraft offers *multimodal* data, not just visual data.

NovelCraft is complementary to other scene-focused datasets as some methods may not want to try to overcome the challenge of NovelCraft's severe class imbalance or aim for collaboration with an agent instead of a more typical classification setting. To our knowledge there are no scene-focused datasets that are class-balanced by default, but datasets larger than NovelCraft could more easily be split into subsets to obtain class balancing without reducing performance significantly. A class balanced setting can be more practical for applications where the deployment environment is more controlled, such as robots deployed in a warehouse. UBnormal (Acsintoae et al., 2022) provides sequences of images for novelty detection that are useful for applications utilizing fixed cameras, such as surveillance imagery processing.

**Vision for open worlds.** Several works share our focus on an open world, presenting scene-focused datasets with test classes not seen in the training set. Rambhatla et al. (2021) present novelty discovery and localization results from the scene-focused COCO-2014 dataset. Liu et al. (2019) explore open long-tailed recognition of not-so-common object categories representing the "long tail" using a repurposed view of the scene-focused Places dataset (Zhou et al., 2018). Lin et al. (2021) introduce the CLEAR benchmark for continual learning on scene-focused internet images from 2004-2014. Other works explore open-world instance segmentation, either on the COCO dataset (Saito et al., 2022) or on images collected from cars on the road (Chan et al., 2021). Compared to these works, our dataset is distinct in its multimodality and how images are gathered over time by a navigating agent.

**Related Minecraft datasets.** Very recent parallel work has developed the MineDojo benchmark framework (Fan et al., 2022), which evaluates MineCraft agents that can learn from vision to perform 32 programmatic tasks like "hunt sheep" or "combat zombie" as well as 32 creative tasks like "build a haunted house", by training on an internet-scale knowledge base of multimedia recordings, tutorials, and discussions. This extends early work on MineRL by Guss et al. (2019), focused on one "obtain diamond" task. Their evaluations focus on the agent's success rate on new tasks in new visual environments. Our evaluations specifically evaluate novelty detection and generalized category discovery, and thus should be seen as complementary to their work. Additionally, our NovelCraft dataset provides symbolic world-state data absent in other datasets.

**Industrial datasets.** Two recent efforts are designed to enable industrial inspection scenarios. The MVTech Anomaly Detection dataset (Bergmann et al., 2019) depicts textures like carpet or glass deliberately altered with *anomalous defects*, such as scratches or dents. Huang & Wei (2019) offer a similar dataset with images of circuit board defects. Unlike these works, our target scenario is an agent navigating open worlds over time, not a fixed camera on an assembly line. Further analysis of other related work can be found in App. D.

## 3 NovelCraft Dataset

### 3.1 Background: PolyCraft Virtual Environment

The virtual world we build upon (with permission from the creators) is PolyCraft[2], a modification of the videogame MineCraft developed by a research team at UT-Dallas (Smaldone et al., 2017; Goss et al., 2023). Polycraft is multi-purpose software with several applications. We use a particular environment, PolyCraft POGO.

Within the Polycraft POGO environment, an agent is tasked with exploring a 3-dimensional voxel world via a sequence of actions such as move, break, or craft. Completing the task requires the agent to execute a plan roughly 60 actions long. Moving the agent any distance requires only a single action, so many actions involve the agent interacting with an object or another environment-controlled agent. The agent's goal is to create a pogo stick from resources, such as wood and rubber, that must be gathered from the environment. A solution to the task requires the agent to break trees to acquire wood, craft the wood into planks and sticks, move to a work bench, use the work bench to craft the planks and sticks into a tree tap, place the tree tap adjacent an unbroken tree, use the tree tap to acquire rubber, move to a work bench, and use the work bench to craft rubber, planks, and sticks into a pogo stick. In the event of a novel change in the environment, such as some trees not producing rubber, the agent must be able to change multiple steps in the solution, such as breaking those trees for wood and not using the tree tap on them, to successfully complete the modified task.

At each step, the agent observes both an image and the environment's symbolic state in JSON text format. Example images and symbolic information are shown in Fig. 2. Each image is a 256x256 pixel RGB depiction of the agent's current perspective. The JSON text includes positions and names for every object in the environment and every environment-controlled agent, as well as the agent's position and state information (what materials it has collected). This representation is compatible with other possible use cases for agent-based open world novelty detection, where the JSON may represent the environment by listing sensor measurements.

### 3.2 Data Collection

Our dataset consists of prerendered observations from an agent pursuing the POGO task, with contents summarized in Tab. 2. The data is organized into episodes, an ordered set of data representing the sequence of steps taken by the agent solving the POGO task. Each episode starts with a different initial environment and ends after the agent has completed the task. Each episode is represented by a folder with both an RGB image and a JSON file detailing the state of the environment after each action the agent took. Each image is labeled, indicating if it depicts a non-standard object.

| Split | Type | Episodes | Num. Classes | | Num. Images | |
|---|---|---|---|---|---|---|
| | | | Object | Gameplay | Raw | Filtered |
| Train | Normal | 165 | 3 | 2 | 9707 | 7037 |
| | Normal+ | 2575 | 0 | 1 | 125636 | 125636 |
| | Novel | 0 | 0 | 0 | 0 | 0 |
| Valid. | Normal | 21 | 3 | 2 | 1154 | 873 |
| | Novel | 50 | 5 | 0 | 2520 | 332 |
| Test | Normal | 21 | 3 | 2 | 1194 | 890 |
| | Novel | 440 | 43 | 1 | 20988 | 3530 |

Table 2: Summary statistics of our NovelCraft dataset. Each episode contains a sequence of images and associated JSONs. Each episode has a class label indicating how the POGO environment was modified. The same 5 Normal classes appear in every split: 2 gameplay classes (standard and fence) and 3 modified-object classes. Novel classes are not repeated between validation and test. The Normal+ training split is additional data from the standard environment added to Normal from our larger NovelCraft+ data.

Each episode has a class label indicating how the POGO environment was modified. We collect images under 3 broad scenarios: the standard environment with no modifications, using creator-designed modifications that impact gameplay, and modifying the environment by inserting new objects that are visible but do not impact

---

[2]https://www.polycraftworld.com

gameplay (standard agent can still solve the task). Only one modification is applied per episode, so there is never more than one novelty type in an episode. Fig. 1 shows visual examples for a subset of modifications, with Fig. A.1 showing visual examples of all modifications and summary statistics shown in Tab. A.2.

For all collection, we used an autonomous agent following the DIARC cognitive architecture (Scheutz et al., 2019) to solve the pogo-stick assembly task, as described in Muhammad et al. (2021). This agent does *not* make use of visual images, relying purely on the symbolic world-state available at each step.

**Standard episodes.** Using the standard environment, we collected a base dataset of 99 standard episodes. Our agent typically produces 45-56 images per episode (5th-95th percentile).

**Gameplay-altering modifications.** The UT-Dallas PolyCraft team created several modifications that would impact gameplay. We selected two that would be visually recognizable: inserting fences that block access to trees and inserting a tree species with distinct bark that can be harvested more efficiently. Harvesting trees is required to build the pogostick, so both modifications modify agent behavior. We collect 80 fence episodes and 85 tree episodes, with our agent producing 29-72 images per episode (5th-95th percentile range).

**Inserted object modifications.** From a library of possible PolyCraft objects that never appear in the standard task, we selected 51 object types (Tab. A.2) and inserted them into the POGO environment. These objects do not assist or hinder the agent attempting the POGO task, so agent behavior is not changed. We collected at least 10 episodes each, with 32-56 images per episode (5th-95th percentile range).

**NovelCraft+: Extra standard episodes.** To facilitate models requiring more data we have released NovelCraft+, containing over 2500 episodes and over 100,000 additional frames of standard environment training data. This significantly increases dataset size but results in exaggerated class imbalance between standard and novel images. We use NovelCraft+ to evaluate visual novelty detection (Sec. 4), where class imbalance is best understood and has the least impact. In Sec. 5 and Sec. 6, the original smaller NovelCraft is used for symbolic and multimodal novelty detection and generalized category discovery.

**Filtering to identify non-standard images.** Within a modified episode, only a few images may actually show non-standard objects. Rendering semantic segmentation maps within the game engine would have been a significant engineering challenge. Therefore we use the locations of agents and objects in the JSON to reconstruct the agent's view within the 3D rendering software Blender (Blender Foundation, 2018). Not all camera parameters were available, so a grid search was used to maximize mean intersection over union for non-standard objects in manually annotated images, achieving 93.0%. For each frame from a modified environment, we can thus render semantic segmentation maps highlighting non-standard objects and record the percentage of pixels containing a non-standard object. We then obtain representative frames from modified episodes by filtering out images with less than 1% non-standard content. This mitigates noise from the rendering, while maintaining challenging examples where non-standard objects are small but clear to a human. Per-frame percentage labels and code for this filtering step are provided in our data release.

**Splitting classes into normal/novel and episodes into train/valid/test.** We consider 5 total class labels as normal. The "Standard" environment with no modifications, the "Gameplay:Fence" gameplay-altering modification, and the "Item:Anvil", "Item:Sand", and "Item:CoalBlock" inserted object modifications. This enables methods that repurpose classifiers of multiple normal classes to perform novelty detection. These classes were chosen for containing a high number of images depicting novelty to ensure a reasonable number of images in the training, validation, and test sets. Choosing classes with fewer images depicting novelty would likely decrease model performance due to a smaller training set.

Each episode of the normal classes is assigned to either training, validation, or test. To compensate for differing episode sizes, we selected a split that best matches an 8:1:1 ratio of *filtered* images assigned to train/validation/test out of 100 possible random splits. We randomly picked 5 additional inserted object classes to represent novelty exclusively in the validation set. Finally, all remaining inserted object classes and gameplay classes represent novel classes exclusively in the test set.

In most past work, novelty or anomaly detectors are informed by only normal examples, with no examples labeled novel in training or validation sets. In contrast, our validation set provides examples from 5 *novel* classes as well as normal ones. We do this because in practice, often there are *some* available examples of

novelties. To not use any such examples to inform models is a missed opportunity. This is similar to the validation set construction of Liang et al. (2018), but provides more challenging novelties (closer appearance to normal). All validation novelties are distinct from test set novelty to avoid leakage. Future users could create other splits as desired to better match their own intended use cases (e.g. no novelties in validation).

## 4   Visual Novelty Detection

Here, we describe how we use NovelCraft+ to formulate and evaluate the task of visual novelty detection (Sec. 4.1), which methods we consider (Sec. 4.2), and the results of our experiments (Sec. 4.3). Appendix Sec. E provides experiments assessing how training data size impacts results (NovelCraft+ vs. NovelCraft).

### 4.1   Task Description and Evaluation Plan

The goal of our visual novelty detection task is to develop methods that can distinguish whether an image is *novel* or not. The predefined training set defines 5 normal classes. The goal on test data is to assess if a given image contains content that is *not* in one of the *normal* classes.

**Performance metrics.** Detectors are evaluated using receiver-operating characteristic (ROC) curves and precision recall (PR) curves. These visualize tradeoffs across possible thresholds that could be applied to the decision score. The ROC curve assesses tradeoffs between true positive rate (TPR, aka recall) and true negative rate (TNR), while PR compares TPR versus positive predictive value (PPV, aka precision). PR curves provide information about *imbalanced* tasks like novelty detection that ROC curves alone cannot (Saito & Rehmsmeier, 2015). To summarize performance, we report the area under both curves. Higher numbers mean better detectors. We bold the best score and any others within 1.0 percentage point.

Measuring the area under ROC and PR curves is insufficient for applications that require automated *binary* decisions of novel or not. Instead of averaging over many thresholds, one specific threshold must be chosen, ideally in a way that respects the task-specific costs of possible mistakes. We consider two different cost regimes when reporting metrics like TPR, TNR, and PPV. First, we prioritize avoiding false negatives, by selecting a threshold achieving 95% TPR. Liang et al. (2018) also evaluated OOD detectors in this regime. Second, we avoid too many false positives among all alerts, via a threshold enforcing 80% precision (PPV). As we later show in Tab. 3, the relative ranking of different detectors can *change* depending on which regime we select. Picking a method based on area-under-the-curve metrics alone might lead to *under-performance* in the applied regime of interest. This critical distinction may be under appreciated in current evaluation practice.

**Frequency of novelty.** The relative frequency of the positive class (novel) matters for metrics such as precision. In a typical novel episode, roughly 25% of frames depict novelty. Yet it is difficult to properly score images where very few pixels (0.01-1%) show the novel object. To handle this, we only score images in the *filtered* subset of the test set, where we are more confident in the label. Then, for metrics where frequencies matter, we take a weighted average with 75% weight on normal images and 25% on novel images.

### 4.2   Methods for Visual Novelty Detection

We consider several detection methods representing a variety of approaches. We follow Cheng & Vasconcelos (2021) in finetuning a **common pretrained VGG-16 network architecture** (Simonyan & Zisserman, 2015) for each method utilizing deep classifiers. Each method can access the predefined validation set to **tune hyperparameters**. For classifiers, grid search maximizes validation accuracy on the normal classes. Novelty detectors then select to maximize validation AUROC. Details for reproducibility (esp. settings for training and hyperparameter grid search) are in the Appendix.

We first consider **NDCC** (Cheng & Vasconcelos, 2021) as a state-of-the-art way to *adapt the training objective* of a deep classifier to handle novelty. NDCC learns class-specific Gaussians in the penultimate-layer feature space, which are used to calculate Mahalanobis distances for an input image at test time. The minimum distance is used as the novelty score, which assumes novel images are embedded farther away from the closest Gaussian than normal images.

We also try detectors that repurpose standard deep classifiers. First, a **Deep Ensemble** (Lakshminarayanan et al., 2017) finetunes 5 distinct VGG-16 networks on normal data, then takes the max of averaged softmax outputs as the novelty score. Second, **ODIN** (Out-of-DIstribution detector for Neural Networks)  (Liang

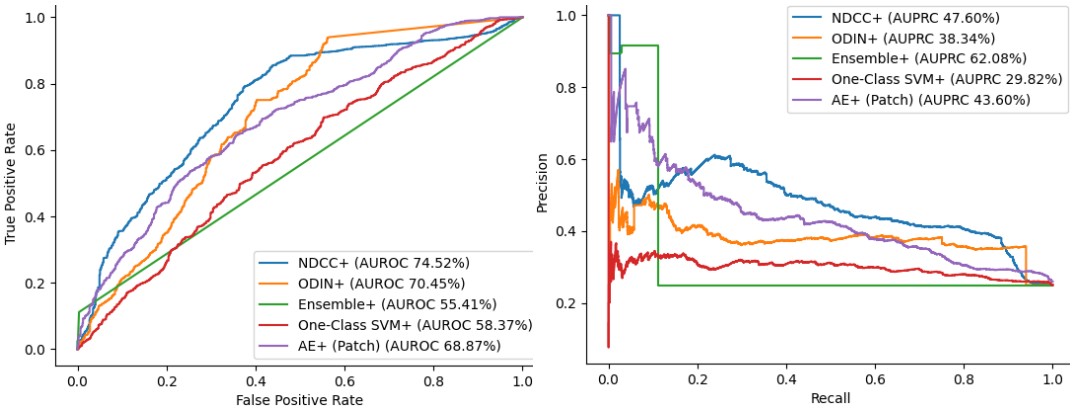

(a) Receiver Operating Characteristic (ROC) curves    (b) Precision Recall curves

Figure 3: Visual novelty detection tradeoff curves (higher is better), evaluated per-frame on NovelCraft test images. The + symbol in a method's legend name denotes that our larger NovelCraft+ dataset was used for training.

| Method | AUROC | AUPRC | at TPR 95% | | at PPV 80% | |
|---|---|---|---|---|---|---|
| | | | TNR | PPV | TNR | TPR |
| NDCC+ | **74.52** | 47.60 | 9.32 | 25.89 | 99.89 | 2.55 |
| ODIN+ | 70.45 | 38.34 | 0.00 | 25.00 | **100.00** | 0.11 |
| Deep Ensemble+ | 55.41 | **62.08** | 0.00 | 25.00 | 99.89 | 2.86 |
| One-Class SVM+ | 58.37 | 29.82 | 9.66 | 25.96 | **100.00** | 0.00 |
| Autoencoder+ | 68.87 | 43.60 | **22.02** | **28.88** | 99.78 | 2.72 |

Table 3: Visual novelty detection performance metrics (percentages, higher is better), evaluated per-frame on test images. NovelCraft+ used for all results here (denoted with +). NDCC performs best in AUROC but is outperformed by Deep Ensemble in terms of AUPRC, and further outperformed by the Autoencoder in the TPR 95% regime.

et al., 2018) creates an adversarial perturbation (Goodfellow et al., 2015), feeds that through the model, and takes the max of the temperature-scaled softmax output as the score.

Next, we consider a denoising **Autoencoder (AE)** trained without labels to encode a normal image then reconstruct it accurately. We borrow encoder and decoder architectures from Abati et al. (2019). At test time, the reconstruction error is used as a novelty score, which assumes novel classes will be reconstructed less accurately than the normal classes (Richter & Roy, 2017). Our dataset has large images (256x256), so we consider reconstructions that process 32x32 *patches* (whole image AEs performed worse). After reconstructing each patch, we aggregate the novelty scores via an average over all patches.

We finally try a **One-Class Support Vector Machine** (**OC-SVM**) (Schölkopf et al., 2000). We use the penultimate layer of a finetuned VGG-16 as features and pick an RBF kernel with tuned lengthscale. This model was fit on a subset of 10,000 images due to the method scaling poorly with dataset size.

### 4.3   Results and Analysis of Visual Novelty Detection Experiments

**Tradeoff curves.** Using our new dataset, we evaluated all detection methods described in Sec. 4.2 and report their ROC and PR curve in Fig. 3. NDCC clearly outperforms other methods in AUROC overall, but is beaten by ODIN and Autoencoder at high FPR. Deep Ensemble and the Autoencoder perform well at very low FPR but are surpassed by NDCC above FPR 5%. The One-Class SVM outperforms Deep Ensemble above FPR 30%. The same trends are seen in the precision recall curves which better visualize the effects of novel classes composing only 25% of the test set. Deep Ensemble and Autoencoder reach higher precision than others at low recall values. NDCC surpasses the Autoencoder at 20% recall and remains the best performing method until it is beaten by ODIN at 90% recall. These results suggest that Deep Ensembles or Autoencoders may be preferred in applications requiring high precision, despite NDCC and ODIN generally being considered state of the art.

**When false negatives matter most.** In Table 3, when enforcing TPR of 95%, the Autoencoder performs best followed by the One-Class SVM and NDCC. However, there is still clearly room for significant improvement as false positives outnumber both true negatives and true positives.

**When false positives matter most.** In Table 3, when enforcing precision of 80%, we find that NDCC, the Autoencoder and Deep Ensemble perform similarly well with respect to their TNR. TPR values are suprisingly low in the 80% precision regime. This suggests an opportunity for models specialized for low false positive regimes (Rath & Hughes, 2022).

## 5 Symbolic and Multimodal Novelty Detection

While Sec. 4 focused exclusively on visual processing, we will now compare vision-only models to symbolic models that consume the JSON world-state available at each frame, as well as *multimodal* models that combine visual and symbolic input. To make this evaluation fair and interesting, here we must focus exclusively on gameplay-altering novelties only, so that the symbolic detection task is non-trivial. In this case, it makes more sense to evaluate novelty detection at the *episode-level* (aggregating across frames observed over time) rather than at the frame-level. Understanding episode-level performance is valuable for assessing agents acting in the same world over time, as we can highlight key properties such as time-delay until correct detection.

### 5.1 Task Description and Evaluation Plan

**Focus on gameplay modification.** In our dataset, most modified environments introduce novel objects into the game world that would be trivial to detect symbolically as a never-before-seen key in the JSON description of the world. However, the *gameplay* novelties by design cause changes in the observable behavior of both agent and environment over time, which a good agent could detect. We thus focus on these gameplay novelties in this task, avoiding trivial easiness by masking any non-standard JSON keys. Because our numerous inserted-object novelties are not appropriate for this task, only a few gameplay novelties remain available. We thus use only the standard episodes from the base NovelCraft dataset for training and validation to maximize diversity in this task's test set, which contains standard episodes and all available gameplay novelty episodes. In addition to the Fence and Tree novelties, we add new Supplier and Thief gameplay novelties, which are exclusive to this task due to only being labeled at the episode level instead of at the frame level. These novelties introduce artificial agents that respectively help and hinder the player, and were released after our initial experiments in Sec. 4 were completed.

**Episode-level predictions.** In our proposed benchmark task, models have access to both the RGB image and the JSON symbolic state at each frame in an episode (vision-only methods will ignore the JSON, and symbolic-only methods will ignore the image). The required output is one novelty score for each *episode*. For simplicity, all models predict novelty scores for individual frames, then to score an episode we take the maximum score over all frames. Other strategies for episode-level modeling can be tried in future work.

**Performance metrics.** We use the same performance metrics as in Sec. 4.1, evaluated on episodes instead of single frames. While novelty may be easier to detect after experiencing all frames in an episode, failing to detect novelties until the very end of an episode compromises an agent's ability to adapt or take advantage of novelty. We thus also report *average detection delay*: the number of contiguous frames from the start before a true novelty is detected, averaged across all novelty-containing episodes.

### 5.2 Methods for Multimodal Novelty Detection

**Symbolic approach.** We reason that jointly modeling the agent's behavior and its environment is useful, as both may be affected by gameplay novelty. We transform the JSON into a multivariate time-series where each frame $t$ is represented by a vector $\mathbf{x}_t$ of 24 numeric and 2 categorical variables selected from JSON entries for the player and the world. We model the time series via an $K$-th order autoregressive model, trained to minimize predictive error over the normal-only training set (Choi et al., 2021; Blázquez-García et al., 2021). At test time, for each frame, the model predicts the current value $\hat{\mathbf{x}}_t$ from a fixed context of the $K$ previous observed frames. The novelty score at frame $t$ is then computed as the error between predicted and observed values: $||\hat{\mathbf{x}}_t - \mathbf{x}_t||$. The architecture of our symbolic detector is a simple multi-layer perceptron (MLP), with two autoregressive context sizes: $K \in \{2, 5\}$. Hyperparameters such as number of layers, hidden dimensions and drop-out rate are tuned for each $K$ to minimize validation error. Reproduciblity details are in the Appendix.

| Model | AUROC | AUPRC | at TPR 95% | | | at PPV 95% | | |
|---|---|---|---|---|---|---|---|---|
| | | | TNR | PPV | Delay | TNR | TPR | Delay |
| Visual (AE Patch) | 81.62 | **95.61** | 31.34 | **88.72** | **5.32** | 80.60 | 66.47 | 14.41 |
| Symbolic (2 step) | 76.12 | 94.18 | 4.48 | 84.96 | 10.39 | **98.51** | 5.04 | 33.24 |
| Symbolic (5 step) | 81.51 | **95.28** | 11.94 | 85.97 | 10.19 | **98.51** | 5.04 | 32.89 |
| Multimodal (2 step) | 80.46 | 94.65 | **34.33** | **89.15** | 7.44 | **98.51** | 5.04 | 33.14 |
| Multimodal (5 step) | **85.17** | **95.83** | **34.33** | **89.15** | 6.93 | 77.61 | **75.07** | **12.72** |

Table 4: ***Episodic*** novelty detection performance (higher is better for all but delay) on gameplay-only test episodes (Sec. 5). Multimodal models perform better than their components with similar delay to visual models. *These per-episode results are not directly comparable with per-frame evaluation in Tab. 3.*

**Multimodal approach.** Our multimodal model is a simple ensemble of the visual and symbolic base detectors. For each base detector, we normalize scores to range from 0 and 1 on the validation set. At test time, the multimodal detector reports the sum of the two normalized scores from visual and symbolic. We chose a patch-based autoencoder for the vision component because it has good performance (top 3 ranking or better in every metric of Tab. 3) and does not require more than one training class (unlike NDCC or ODIN).

### 5.3 Results and Analysis: Symbolic and Multimodal Novelty Detection

Tab. 4 reports the performance of visual, symbolic, and multimodal detectors. Overall, this episode-level evaluation intuitively produces higher TPR, TNR, and PPV scores than previous frame-level analysis, due to less sensitivity to frame-level noise. Looking at symbolic performance, 5-frame contexts yield significantly better performance than 2-frame contexts on ROC based metrics, suggesting medium-range temporal modeling matters for our benchmark. Finally, the multimodal ensemble nicely blends the best of both visual and symbolic models to improve performance with delay similar to the visual model across both cost regimes. Future methods have plenty of room to improve (human performance would be close to 100% AUROC).

## 6 Generalized Category Discovery

### 6.1 Task Description and Evaluation Plan

Both *generalized category discovery* (GCD) (Vaze et al., 2022) and novel category discovery (NCD) (Zhao & Han, 2021; Han et al., 2020) consider the problem of assigning labels to a set of unlabeled data by utilizing a set of related labeled data. NCD assumes this unlabeled data contains only new classes not seen in the labeled data, while GCD allows labeled classes and new classes to appear in the unlabeled set. Unlike novelty detection, both the labeled and unlabeled data are available during training. During testing, clustering accuracy is used to evaluate the labels assigned to the unlabeled set. We additionally report clustering accuracy on the subset of classes seen in the labeled set as well as the new classes.

To try GCD on our base NovelCraft data, we utilize our training split as the labeled set and our validation split as the unlabeled set. This results in 5 classes in the labeled set and 10 classes in the unlabeled set, which includes new examples of the labeled set classes. In our evaluation we assume prior knowledge of the number of new classes in the unlabeled set, but future evaluations may utilize the method in Vaze et al. (2022) to select the number of new classes. We leave GCD assessment on our larger NovelCraft+ to future work, as methods for GCD under severe class imbalance have not been well explored to our knowledge.

### 6.2 Methods for Category Discovery

**GCD SSKM.** Following Vaze et al. (2022), we use a ViT-B-16 (Dosovitskiy et al., 2020) vision transformer pretrained with DINO (Caron et al., 2021) self-supervision on unlabeled ImageNet then *fine-tuned* with the GCD contrastive loss on NovelCraft. The contrastive loss proposed by Vaze et al. (2022) has an unlabeled term and a labeled term, with the unlabeled term applied to all data and the labeled term applied only to labeled data. The unlabeled term encourages two random augmentations of the same image to be mapped close to each other in embedding space, penalizing being close to different images in the same mini-batch. The labeled term encourages each pair of images that share a class to be mapped close to each other in embedding space, penalizing being close to images from other classes.

Class-balanced sampling is utilized on the labeled set to account for class imbalance. The learned features from this model are then fed into semi-supervised k-means (SSKM) (Vaze et al., 2022), which learns cluster assignments for all examples that respect known labels from the labeled set.

**DINO SSKM baseline.** To examine the effectiveness of GCD's fine-tuning, we compare to a semi-supervised k-means baseline that uses pretrained DINO features. We chose this instead of unsupervised k-means as in Vaze et al. (2022), so that both methods utilize the labeled set examples.

**Semi-supervised Gaussian mixture.** To examine the effectiveness of SSKM's clustering, we compare to a semi-supervised Gaussian mixture model (SSGMM). SSGMM is a new extension of Gaussian mixture models that applies the semi-supervision method of SSKM (Vaze et al., 2022) to learn cluster assignments that respect known labels from the labeled set. SSGMM should be more flexible as it allows soft assignment (multiple clusters can take partial responsibility for a given data example), can learn the frequencies of each cluster, and can learn the appropriate scaling of a cluster-specific spherical covariance. In contrast, k-means assumes hard assignment (each data example matched to exactly one cluster), assumes uniform cluster frequencies, and represents the limiting behavior when all clusters share a single spherical covariance whose scale diminishes to zero. We use a modified version of expectation maximization (Dempster et al., 1977) to fit this GMM in semi-supervised fashion. Semi-supervision is introduced by modifying the expectation step so that each labeled example is assigned to the correct cluster corresponding to its label as in SSKM, with no responsibility assigned to incorrect clusters.

### 6.3 Results and Analysis

Fig. 4 provides quantitative results from our generalized category discovery experiments. The features learned during GCD training clearly improve clustering accuracy, with most of the increases coming from the labeled set. The choice of semi-supervised clustering method has little effect after GCD training, but SSGMM improves labeled accuracy over SSKM using pretrained DINO features.

Comparing to results on classification datasets in Vaze et al. (2022), NovelCraft is found to be far more challenging than CIFAR-10 (Krizhevsky, 2009), the only other dataset tested with comparable class splits. Comparable loss in accuracy between labeled and new classes is only seen in experiments with hundreds of classes on fine-grained classification datasets.

The confusion matrix reveals two ways in which class imbalance and scene-focused images, not present in previously tested classification datasets, significantly reduces accuracy. In the labeled classes, we see that the small labeled item classes (like I3) are clustered with the more frequent new item classes (I5), despite labels informing the semi-supervised k-means step. Additionally, many of the new classes are clustered with the standard class. The lack of confusion with the labeled gameplay novelty class, which has similar frequency to the standard class, suggests an opportunity for learned features to distinguish novelty from the standard class specifically.

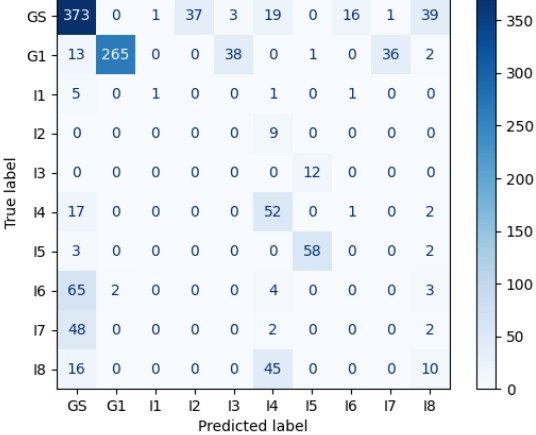

| Method | Clustering Accuracy | | |
|---|---|---|---|
| | All | Labeled | New |
| GCD SSKM | **72.3** | **83.9** | **41.9** |
| GCD SSGMM | 71.5 | 83.7 | 39.1 |
| DINO SSKM | 30.5 | 32.4 | 25.7 |
| DINO SSGMM | 34.8 | 38.3 | 25.5 |

Figure 4: ***Visual category discovery*** results. *Top:* Confusion matrix for GCD SSKM. The labeled set includes gameplay classes (GS, G1) and 3 item modifications (I1, I2, I3). Remaining classes are new item modifications. *Bottom:* Table of clustering accuracy (percentages) for all classes, labeled classes, and new classes (only in the unlabeled set). We report the mean accuracy across 5 clustering fits from random initializations, using the same neural representations.

Improving GCD to address class imbalance in unlabeled data will be challenging. Balancing the labeled examples used by semi-supervised k-means risks increasing the splitting of more frequent classes, such as the predicted I3 and I4 classes consisting of a subset of the most frequent classes. Separating new classes from the standard class likely requires specialized modifications to the loss, as all unlabeled data resemble standard class images to some extent.

## 7 Discussion and Outlook

We hope our NovelCraft dataset inspires new methods development on novelty detection and characterization, enabling agents to explore open worlds.

**Limitations.** Our benchmark focuses on one specific open world. We wish to avoid overclaiming about generality (Raji et al., 2021). Many other open worlds are possible. The kinds of visual properties we call *novel* emphasize object types. We could instead have varied counts, sizes, spatial relationships, or co-occurrance patterns between objects. We include only a few gameplay modifications due to the cost of creating these. More diverse gameplay novelties that are not trivial to detect symbolically would be worth future investment. Our multimodal experiments are limited to our specific choice for JSON data. JSON can encode less precise sensory inputs which are worth further exploration.

**Future Directions.** We can foresee several promising lines of work building on our NovelCraft data:

*Multimodal novelty detection.* A recent review suggests that multimodal novelty detection is "a largely unexplored research area" (Pang et al., 2022). Our dataset provides two complementary modalities, symbolic and visual, that could both be leveraged by future methods.

*Continual learning.* Our new dataset is naturally episodic in nature. Future work could use a *continual learning* paradigm (Lin et al., 2021) instead of "train once, then deploy", which is far more realistic for open-world learning. Our discovery task in Sec. 6 could be a promising first step.

*From vision to action.* Our dataset is collected by a real artificial agent interacting in a virtual world. Future work could use visual novelty processing to inform how the agent selects actions to solve its task. *Active perception* is especially promising: novel objects may or may not look novel from all perspectives; an agent may need to actively obtain distinct perspectives to determine a new object's properties and how it could be used in service of the agent's goal.

### Broader Impact Statement

We have designed our dataset with the goal of positive benefit to society. We think novelty detection and characterization can help build agents that recognize when a model is not suitable (and might even be harmful) and adapt accordingly. We hope that our findings on the difficulties of adapting models from research settings to applications encourages users to *always* inspect a model's potential impacts. We recognize that there is a potential for misuse, as with all classifiers and especially visual ones, if the classifier is used to discriminate by gender, ethnicity, and other protected categories. Ultimately, we hope that the potential of novelty detection to preempt accidental misuse will outweigh the impacts of purposeful misuse of these techniques.

### Acknowledgments

This work was primarily funded by the Defense Advanced Research Projects Agency (DARPA), a research and development agency of the United States Department of Defense, as part of the Science of Artificial Intelligence and Learning for Open-world Novelty (SAIL-ON) program under grant W911NF-20-2-0006. Author SS reports funding from the Austrian Institute of Technology via the Decision Making and Cognitive Control of Complex Systems program.

We thank the members of the PolyCraft team at UT Dallas, especially Dr. Eric Kildebeck, Stephen Goss, and Dr. Walter Voit, for creating an interesting platform for novelty research, approving the release of images for this dataset, and providing support throughout the DARPA SAIL-ON program.

At Tufts University, we thank Evan Krause and Ravenna Thielstrom for their work in developing the DIARC agent for Polycraft.

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

## Appendix Contents

## A    Dataset Documentation

### A.1   Links to Data and Code

**Dataset Website:** https://novelcraft.cs.tufts.edu/

**Data Download:** https://tufts.box.com/shared/static/fq0awbrahmsr97zetqo1v2uz5rjkvon6.zip

**Dataset Code:** https://github.com/tufts-ai-robotics-group/polycraft-novelty-data/tree/TMLR

**Methods Code:** https://github.com/tufts-ai-robotics-group/polycraft-novelty-detection/tree/TMLR

**GCD Datasets Code:** https://github.com/tufts-ai-robotics-group/GCDdatasets

### A.2   Licensing

The dataset is released under a CC-BY license and the associated code is released under a MIT License.

### A.3   Hosting and Maintenance

The dataset will be hosted by Tufts University through their cloud service provider Box.com. Tufts is planning on using this service indefinitely, but the dataset will be transferred and links updated if a provider change does occur.

Additionally, the Tufts University Computer Science Department will host a website with documentation, dataset download links, and backups of all externally hosted files. This is supported by the department's dedicated IT team, which ensures server maintenance and backups. This website and server will be maintained indefinitely.

### A.4   Statistics and Examples

See Fig. A.1 on the next page for an example image from every possible novelty type.

See Table A.2 for statistics summarizing the number of *filtered* episodes and images available for every single novelty type.

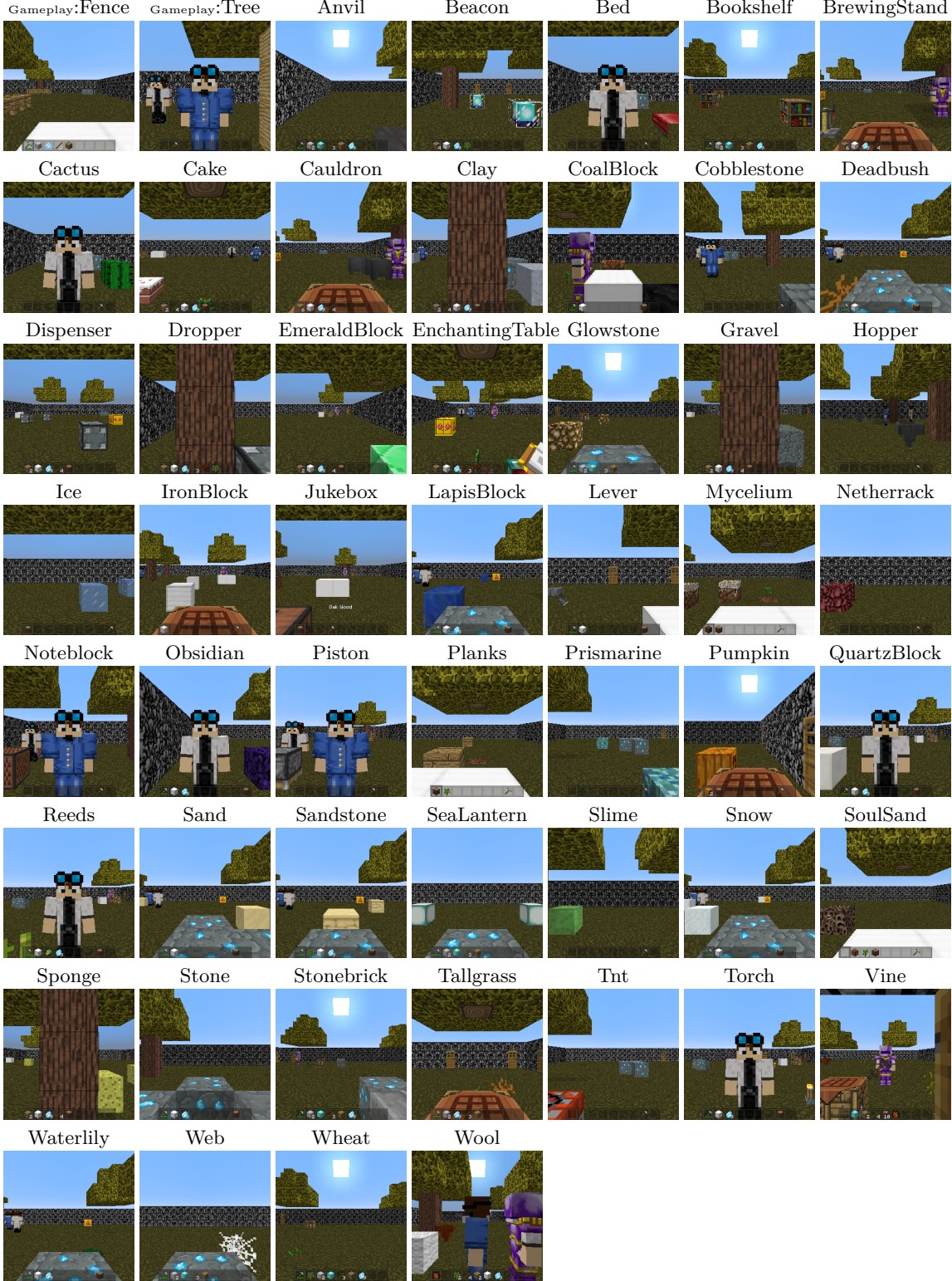

Figure A.1: Example image for every novelty in NovelCraft.

| Modification Name | Modified Episodes | Total Filtered Frames |
|---|---|---|
| Gameplay:Fence | 80 | 3581 |
| Gameplay:Tree | 85 | 619 |
| Item:Anvil | 10 | 97 |
| Item:Beacon | 10 | 55 |
| Item:Bed | 10 | 81 |
| Item:Bedrock | 10 | 420 |
| Item:Bookshelf | 10 | 77 |
| Item:BrewingStand | 10 | 91 |
| Item:Cactus | 10 | 89 |
| Item:Cake | 10 | 66 |
| Item:Cauldron | 9 | 68 |
| Item:Clay | 9 | 68 |
| Item:CoalBlock | 8 | 109 |
| Item:Cobblestone | 9 | 72 |
| Item:Deadbush | 6 | 33 |
| Item:Dispenser | 8 | 85 |
| Item:Dropper | 9 | 57 |
| Item:EmeraldBlock | 9 | 82 |
| Item:EnchantingTable | 10 | 84 |
| Item:Glowstone | 8 | 45 |
| Item:Gravel | 10 | 59 |
| Item:Hopper | 10 | 77 |
| Item:Ice | 9 | 75 |
| Item:IronBlock | 9 | 80 |
| Item:Jukebox | 10 | 54 |
| Item:LapisBlock | 9 | 79 |
| Item:Lever | 10 | 61 |
| Item:Mycelium | 10 | 98 |
| Item:Netherrack | 9 | 79 |
| Item:Noteblock | 9 | 71 |
| Item:Obsidian | 8 | 63 |
| Item:Piston | 10 | 83 |
| Item:Planks | 10 | 91 |
| Item:Prismarine | 8 | 74 |
| Item:Pumpkin | 10 | 64 |
| Item:QuartzBlock | 9 | 72 |
| Item:Reeds | 8 | 32 |
| Item:Sand | 10 | 116 |
| Item:Sandstone | 9 | 53 |
| Item:SeaLantern | 9 | 71 |
| Item:Slime | 8 | 56 |
| Item:Snow | 10 | 93 |
| Item:SoulSand | 9 | 90 |
| Item:Sponge | 9 | 54 |
| Item:Stone | 8 | 43 |
| Item:Stonebrick | 8 | 56 |
| Item:Tallgrass | 7 | 76 |
| Item:TNT | 8 | 52 |
| Item:Torch | 10 | 84 |
| Item:Vine | 9 | 73 |
| Item:Waterlily | 7 | 18 |
| Item:Web | 9 | 61 |
| Item:Wheat | 7 | 19 |
| Item:Wool | 10 | 79 |

Table A.2: Count of available episodes and frames available after filtering so at least 1% pixels show novelty.

## B   Dataset Nutrition Label

Dataset nutrition label for NovelCraft dataset, following the suggested format from Holland et al. (2018).

### B.1   Metadata

**Filename** NovelCraft

**File Format** zip

**URL** https://novelcraft.cs.tufts.edu/

**Domain** video game

**Keywords** novelty detection, anomaly detection, generalized category discovery, multimodal, computer vision, open world

**Type** image (PNG), symbolic (JSON)

**Dataset Size** 4.5 GB

**Missing** 0%

**License** CC BY 4.0

**Release Date** June 2022

**Description** Scene-focused, multimodal, episodic data of the images and symbolic world-states seen by an agent completing a pogo-stick assembly task within a video game world. Classes consist of episodes with novel objects inserted. A subset of these novel objects can impact gameplay and agent behavior. Novelty objects can vary in size, position, and occlusion within the images.

### B.2   Provenance

**Environment Source** The Polycraft mod was created by Ronald A. Smaldone, Christina M. Thompson, Monica Evans, and Walter Voit and is available at:

https://www.polycraftworld.com/

The environments used are available at:

https://github.com/StephenGss/PAL/tree/release_2.0

Additional details are provided in Goss et al. (2023).

**Agent Source** The DIARC architecture was produced by Matthias Scheutz, Thomas Williams, Evan Krause, Bradley Oosterveld, Vasanth Sarathy, and Tyler Frasca. The Polycraft agent was produced by Evan Krause and Ravenna Thielstrom.

**Dataset Authors** NovelCraft was produced by Patrick Feeney, Sarah Schneider, Panagiotis Lymperopoulos, Liping Liu, Matthias Scheutz, and Michael C. Hughes and is available at:

https://novelcraft.cs.tufts.edu/

### B.3   Variables

**ID** is composed of the episode type, episode ID, and frame ID. The first two parts of the ID are specified by string and integer folder names. The frame ID is specified by the integer file name.

**Image** is a 256 by 256 pixel RGB image associated with an ID.

**JSON** is a symbolic representation of the environment's state. This includes "player" data, "entities" other than the player, and "map" data detailing the position and properties of objects.

**Target** is provided with the dataset code as "targets.csv" and lists the percentage of novel pixels for each image. Images with target values less than 1% are discarded while others are given the appropriate run type label.

**Splits** is provided with the dataset code as "splits.csv" and lists the splits for the training set classes, divided by episode. Which classes are assigned to novel validation and novel test are listed in "data_const.py".

## C   Training and Model Details

### C.1   Visual Novelty Detection

Except for NDCC and One-Class SVM, all methods are implemented in PyTorch (Paszke et al., 2019) and are trained and tested on a GPU (NVIDIA TITAN X Pascal). NDCC was trained and tested on a NVIDIA RTX 2080 GPU, but note that fewer training epochs is the primary reason for reduced training time. The One-Class SVM is implemented using the SciKit-Learn software package (Pedregosa et al., 2011) and is trained and tested on a CPU. See Table C.1 for training times.

|  | *Training Time* | |
| --- | --- | --- |
| Method | NovelCraft | NovelCraft+ |
| NDCC | 0d 00h 29m 18s | 0d 08h 33m 35s |
| VGG Backbone | 1d 04h 51m 50s | 19d 10h 23m 40s |
| ODIN | 1d 04h 51m 50s [†] | 19d 10h 23m 40s [†] |
| Deep Ensemble | 5d 23h 25m 25s [‡] | 96d 10h 13m 15s [‡] |
| One-Class SVM | 1d 09h 28m 25s [†] | 21d 07h 21m 05s [†] |
| Autoencoder | 0d 20h 01m 53s | 8d 01h 41m 24s |

Table C.1: Training time for the implemented *Visual* novelty detection methods for NovelCraft and NovelCraft+. Results denoted with [†] include the training time for one VGG-16 model. Results denoted with [‡] include the time needed to train 5 separate VGG-16 models, though we note this could be parallelized if sufficient hardware was available.

#### C.1.1   VGG-16 Backbone

A VGG-16  (Simonyan & Zisserman, 2015) is used as a backbone for deep ensemble and ODIN and as a feature extractor for the one-class SVM.

**Architecture.** The VGG-16 model of the PyTorch framework (Paszke et al., 2019) pretrained on ImageNet (Deng et al., 2009) is adapted and fine tuned to classify the normal classes. The pretrained classification head is replaced with a randomly initialized 5 output fully connected layer classication head.

**Training Details.** We optimize the parameters using the cross-entropy loss function and the Adam optimizer (Kingma & Ba, 2014). The "Minecraft item bar" is removed by cropping 22 pixel rows at the bottom of the images. The image intensity values are first scaled to $[0, 1]$ and then normalized by mean values 0.485, 0.456, 0.406 and standard deviations 0.229, 0.224, 0.225. Mean accuracy on the NovelCraft+ training set 98.0%.

**Hyperparameter Tuning.** We tried batch sizes of 16, 32 and 64 images for the VGG-16 classifier fine tuning and found batch size 16 to perform best in terms of per-normal-class accuracy on the validation set. The VGG-16 classifier is trained for 1000 epochs. The learning rate is set to $10^{-5}$, all other parameters of the Adam optimizer are set to the default values of PyTorch (0 weight decay and running average coefficients $\beta_{1,2} = (0.9, 0.999)$). Mean accuracy on the validation set is 96.9%.

#### C.1.2   Deep Ensemble

**Architecture.** The deep ensemble consists of 5 distinct VGG-16 classifiers. The architecture of each VGG-16 classifier models is structured as described in C.1.1, just using different random initializations in the classification head.

#### C.1.3   NDCC

**Architecture and Training Details.** The architecture and training follow the work of Cheng & Vasconcelos (2021), with the following changes to hyperparameters. Accuracy on the NovelCraft+ training set is 90.5%.

**Hyperparameter Tuning.** Each set of hyperparameters used in the supplement of Cheng & Vasconcelos (2021) was tested, with the Stanford Dogs hyperparameters found to work best. To fine-tune further, the Stanford Dogs hyperparameters were modified with all learning rates multiplied by 0.1 or 0.01. The modification with learning rates multiplied by 0.1 was found to perform the best on the validation set. Accuracy on the validation set is 78.5%.

### C.1.4  ODIN

**Architecture.** The architecture of the VGG-16 classifier used for Odin is described in C.1.1.

**Hyperparameter Tuning.** The temperature scaling parameter $T$ and perturbation magnitude $\epsilon$ are set based on a grid search. $T$ is searched for in $[1, 2, 5, 10, 20, 50, 100, 200, 500, 1000]$. $\epsilon$ is searched for in a linearly spaced interval $[0, 0.004]$ using steps of 0.0002. The perturbation magnitude $\epsilon$ leading to highest AUROC on the validation set is 0, $T$ is set to 1000.

### C.1.5  Autoencoder

**Architecture.** The encoder and decoder are from Abati et al. (2019).

**Training Details.** The parameters of the autoencoder are optimized by minimizing the mean squared error (MSE) between the input data and its reconstruction using the Adam optimizer (Kingma & Ba, 2014). The "Minecraft item bar" is removed by cropping 22 pixel rows at the bottom of the images. For the patch-based autoencoder, patches of size $32 \times 32$ pixels are sampled randomly from the cropped images and normalized between 0 and 1. For the full-image-based autoencoder, the lowermost 22 pixel rows are padded and the resulting images normalized between 0 and 1. Gaussian noise with a standard deviation of $\frac{1}{40}$ is added to the input data. Mean reconstruction error on the training set is 0.165.

**Hyperparameter Tuning.** We trained autoencoders with latent representation dimension 50, 100 and 200 and batch sizes 32, 64 and 128. The dimension of the latent representation of the results given is 100 and the batch size is set to 128. The autoencoder is trained for 8000 epochs with a learning rate of $10^{-3}$, all other parameters of the optimizer are set to the PyTorch default values (0 weight decay and running average coefficients $\beta_{1,2} = (0.9, 0.999)$). Mean reconstruction error on the validation set is 0.245.

### C.1.6  One-Class Support Vector Machine

**Architecture.** The one-class support vector machine is implemented based on the OneClassSVM class of the SciKit-Learn software package (Pedregosa et al., 2011). A radial basis function (RBF) kernel is used.

**Training Details.** The One-Class SVM is trained on feature vectors extracted by the fine tuned VGG-16 classifier with its classification head removed (see C.1.1 for architecture and training details). Each feature vector has dimension 4096 and is normalized between 0 and 1.

**Hyperparameter Tuning.** We performed a grid search for *nu* and *gamma* of the One-Class SVM. A suitable value for *nu*, which represents an upper bound on the fraction of training errors and a lower bound on the fraction of support vectors, is searched in $[0.1, 0.2, 0.3, 0.4, 0.5, 0.6, 0.7, 0.8, 0.9]$. The inverse of the length scale of the RBF kernel *gamma* is searched in 9 values logarithmically spaced from $10^{-5}$ to 1. The chosen values performing best in terms of AUROC on the validation set are $nu = 0.8$, $gamma = 10^{-5}$ for NovelCraft and $gamma = 0.00177827941$ for NovelCraft+. The default values of the SciKit-Learn software package are chosen for all other parameters. The One-Class SVM fit is computed on all normal images for NovelCraft. For the NovelCraft+ experiments, the fit is computed on a subset of 10000 images of the normal training data.

### C.2  Symbolic Novelty Detection

**Architecture.** As described in the main paper, the predictive models for symbolic novelty detection consist of multi-layer perceptrons. We use one-hot encoding for categorical variables.

**JSON Feature Extraction.** To represent the game state as a vector, we extract 22 numerical variables which correspond to the inventory count for each of the possible objects that the agent can interact with in

the standard environment. Additionally we extract 2 categorical variables corresponding to the object the agent is currently holding and the object in-front of the agent.

**Training Details.** We train the predictive models using the Adam optimizer with a learning rate of $10^{-4}$ for 50 epochs with early stopping. We use default parameters of 0 weight decay and $\beta_{1,2} = (0.9, 0.999)$ in the optimizer. For numerical features, the loss function optimized is MSE whereas for categorical features we optimize Categorical Cross-Entropy.

**Hyperparameter Tuning.** For our predictive MLP models we fine-tune the number of hidden layers, and a single value for all hidden dimensions. We perform grid search over 1 to 3 hidden layers with step size 1 with hidden dimensions varying between 16 to 64 with step size 16. We perform model selection based on the validation predictive loss. For all three settings of the context size, the final hyperparameters are 2 hidden layers with hidden dimension of 64. We also use a default dropout rate of 0.2 for all models and ReLU activations in all layers except the last.

**Anomaly Scores.** To calculate anomaly scores for a single time-step we take the maximum over the predictive loss of all variables in the vector.

### C.3   Generalized Category Discovery

**Architecture.** Following Vaze et al. (2022), we use a ViT-B-16 (Dosovitskiy et al., 2020) vision transformer pretrained with DINO (Caron et al., 2021) self-supervision on unlabeled ImageNet then *fine-tuned* on NovelCraft.

**Training Details.** Fine-tuning is done with the GCD contrastive loss with supervised loss weight 0.35. Training consists of 200 epochs of SGD optimization with a cosine annealed learning rate ending at the initial learning rate times 0.01.

**Hyperparameter Tuning.** Supervised loss weights of 0 and 1 were tested to verify that the combination of losses was increasing performance. Initial learning rates 1, 0.1, 0.01, and 0.001 were tested with 0.1 found to best minimize training loss, reaching $-1.21$ mean loss.

## D   Further Comparisons to Related Work

### D.1   Further Discussion and Analysis of Related Datasets.

**Surveillance-focused datasets.** Work on anomaly detection in complex scenes has often been motivated by surveillance applications (Ramachandra et al., 2022), such as the UCSD Anomaly Detection dataset (Li et al., 2014). The yearly PETS challenges (Patino et al., 2017) have offered open datasets with anomalies such as abandoned baggage. Unlike these fixed camera datasets, our dataset uses a dynamic, egocentric camera and enjoys fewer ethical concerns than surveillance.

**Comparison to scene-focused datasets.** In Tab. D.1 (next page), we provide a comprehensive comparison of available datasets focused on scene-focused visual tasks that seem especially relevant for novelty/anomaly detection. We thank anonymous reviewers for bringing some of these to our attention.

To recap the major takeaway messages from this table, we suggest that our NovelCraft is the only one containing egocentric images from a goal-oriented agent exploring a 3D world over time. NovelCraft supports benchmark tasks for both detection and category discovery with 40+ classes, while UBnormal, Fishyscapes, and RoadAnomaly datasets offer only a limited number of object classes to be discovered ($<10$). Finally, NovelCraft offers *multimodal* data, not just visual data.

| Dataset | Motivation | Built for novelty / anomaly? | Agent pursuing goal? | Images over time? | Multi-modal? | Future potential for vision-informed actions | Num object classes for detection/discovery? | Num images (train/val/test) |
|---|---|---|---|---|---|---|---|---|
| NovelCraft | Agent building pogostick in Minecraft-like 3D open world | YES | YES | YES | YES | YES | 51 inserted items + 2 gameplay items | 7037 / 1205 / 4420 >130k extra in NovelCraft+ |
| Places [url] Zhou et al. (2018) | Scene recognition: photos covering nearly all possible scenes on Earth | No | No | No | No | No | Object/item labels unavailable | > 1.8 million |
| COCO [url] Lin et al. (2014) | Object detection and instance segmentation in complex scenes from Flickr | No | No | No | YES captions | No | 91 objects Example analysis: (Rambhatla et al., 2021) | 82783 / 40504 / 40775 images |
| UBnormal [url] Acsintoae et al. (2022) | Anomaly detection in artificial videos of complex scenes | YES | No | YES | No | No | 5 object types | 116087 / 28175 / 92640 images from 543 videos |
| Fishyscapes [url] Blum et al. (2021) | Semantic segmentation of anomaly in urban driving scenes | YES | No | No | No | Not easily; would need driving simulator | 10 object types | 2975 / 30 / 1000 |
| RoadAnomaly '19 [url] Lis et al. (2019) | Semantic segmentation of anomalies in street scenes | YES | No | No | No | No | | 60 images |
| RoadAnomaly '21 [url] Chan et al. (2021) | Semantic segmentation of real road hazards | YES | No | No | No | No | 26 object types | 100 images |
| RoadObstacle '21 [url] Chan et al. (2021) | Semantic segmentation of real hazards on road | YES | No | No | No | No | 31 object types | 327 images |

Table D.1: Landscape of available datasets where images focus on complex scenes, not individual objects.

### D.2 Performance Comparion Across Datasets for Visual Novelty Detection

See Table D.2 for visual novelty detection results on other datasets. Our benchmark is more challenging than many existing visual novelty detection evaluations, with only the experiments by Cheng & Vasconcelos (2021) on CUB-200-2010 and Abati et al. (2019) on CIFAR10 getting similar AUROC metrics. We also note that varying model backbones, evaluation methodologies, datasets, and the use of only AUROC as a metric hinders model comparison. We hope to enable future comparisons between models by providing a dataset with defined visual novelty evaluation methodology, evaluations of a variety of methods with the same backbone, and more metrics for detailed model comparisons.

| Method | Backbone Model | In/Out Distribution Dataset | AUROC |
|---|---|---|---|
| NDCC | VGG-16 | **NovelCraft** | 80.6 |
| NDCC | VGG-16 | Stanford Dogs | 92.3 |
| Cheng & Vasconcelos | | FounderType-200 | 94.0 |
| | | CUB-200-2010 | 77.5 |
| | | Caltech-256 | 89.5 |
| ODIN | VGG-16 | **NovelCraft** | 65.5 |
| ODIN | Dense-BC | CIFAR100/TinyImageNet crop | 94.5 |
| Liang et al. | | CIFAR100/TinyImageNet resize | 85.5 |
| | | CIFAR100/LSUN crop | 96.6 |
| | | CIFAR100/LSUN resize | 86.0 |
| Deep Ensemble | VGG-16 | **NovelCraft** | 58.9 |
| Deep Ensemble | WideResNet 28x10 | CIFAR10/SVHN | 96.0 |
| Berger et al. | | CheXpert | 70.4 |
| OCSVM | VGG-16 | **NovelCraft** | 57.7 |
| OCSVM | VGG-F | MNIST | 76.6 |
| Andrews et al. | | X-RAY | 98.7 |
| OCSVM | PCA-Whitening | MNIST | 95.1 |
| Abati et al. | | CIFAR10 | 58.6 |
| Autoencoder | LSA Denoising | **NovelCraft** | 57.5 |
| Autoencoder | LSA Denoising | MNIST | 94.2 |
| Abati et al. | | CIFAR10 | 59.0 |
| | LSA Variational | MNIST | 96.9 |
| | | CIFAR10 | 58.6 |

Table D.2: Visual novelty detection using the tested methods on other datasets. In distribution dataset describes the training dataset and out of distribution dataset describes the dataset used for novel examples. Subsets of the dataset are used when only a single dataset is listed. Evaluation procedures vary so directly comparing results without further context may be misleading. Refer to the cited papers for more details on their specific evaluation.

## E  Further Analysis of NovelCraft+: Extra Train Set of Standard Images

### E.1  Supplementary Data Description

NovelCraft+ consists of supplemental training data for NovelCraft. NovelCraft+ is available through the dataset website or direct download (link redacted). It contains 125,636 additional frames of data from the standard environment. NovelCraft+ is most useful for experiments needing severe class imbalance or that only want to use additional data from the standard environment for training.

### E.2  Results for Visual Novelty Detection on NovelCraft+

Fig. E.2 and Tab. E.1 compare the visual novelty detection methods trained on NovelCraftwith and without the addition of NovelCraft+. The One-Class SVM fit was computed on a subset of 10,000 images for NovelCraft+ due to the method scaling poorly with dataset size. The patch-based autoencoder, the One-Class SVM and Odin gain in novelty detection performance when trained with NovelCraft+ in terms of both AUROC and AUPRC. For the deep ensemble novelty detection, AUROC is decreased when trained on NovelCraft+ while the AUPRC increases. NDCC trained with NovelCraft+ performs slightly worse in both

| Method | AUROC | AUPRC | at TPR 95% | | at PPV 80% | |
| --- | --- | --- | --- | --- | --- | --- |
| | | | TNR | PPV | TNR | TPR |
| NDCC | **80.58** | 49.82 | **47.30** | **37.56** | 99.78 | 2.75 |
| NDCC+ | 74.52 | 47.60 | 9.32 | 25.89 | **99.89** | 2.55 |
| ODIN | 65.54 | 30.35 | 40.56 | 34.77 | **100.00** | 0.14 |
| ODIN+ | 70.45 | 38.34 | 0.00 | 25.00 | **100.00** | 0.11 |
| Deep Ensemble | 58.89 | 41.36 | 0.00 | 25.00 | 99.55 | **5.86** |
| Deep Ensemble+ | 55.41 | **62.08** | 0.00 | 25.00 | **99.89** | 2.86 |
| One-Class SVM | 57.67 | 27.81 | 0.00 | 25.00 | **100.00** | 0.00 |
| One-Class SVM+ | 58.37 | 29.82 | 9.66 | 25.96 | **100.00** | 0.00 |
| Autoencoder (patch) | 57.53 | 38.90 | 9.66 | 25.97 | 99.55 | **5.41** |
| Autoencoder+ (patch) | 68.87 | 43.6 | 22.02 | 28.88 | 99.78 | 2.72 |

Table E.1: Visual novelty detection performance metrics (percentages, higher is better), evaluated per-frame on test images. For the given methods denoted with a +, NovelCraft+ was used for training. While One-Class SVM and autoencoder perform better in terms of AUROC, AUPRC, TNR and precision in the 95% TPR regime, NDCC performs slightly worse in all given metrics except for its TNR in the 80% precision regime. Deep Ensemble increases in AUPRC and TNR, but decreases in AUROC and TPR.

AUROC and AUPRC metrics despite class-balanced sampling during training and longer training times due to increased epoch size. We hypothesize that the decreased performance results from class-balanced sampling insufficiently addressing the severe class imbalance or significantly larger batch sizes being necessary to effectively learn from the larger dataset.

Future work with NovelCraft+ could examine methods to address the severe class imbalance or explore one-class novelty detection methods.

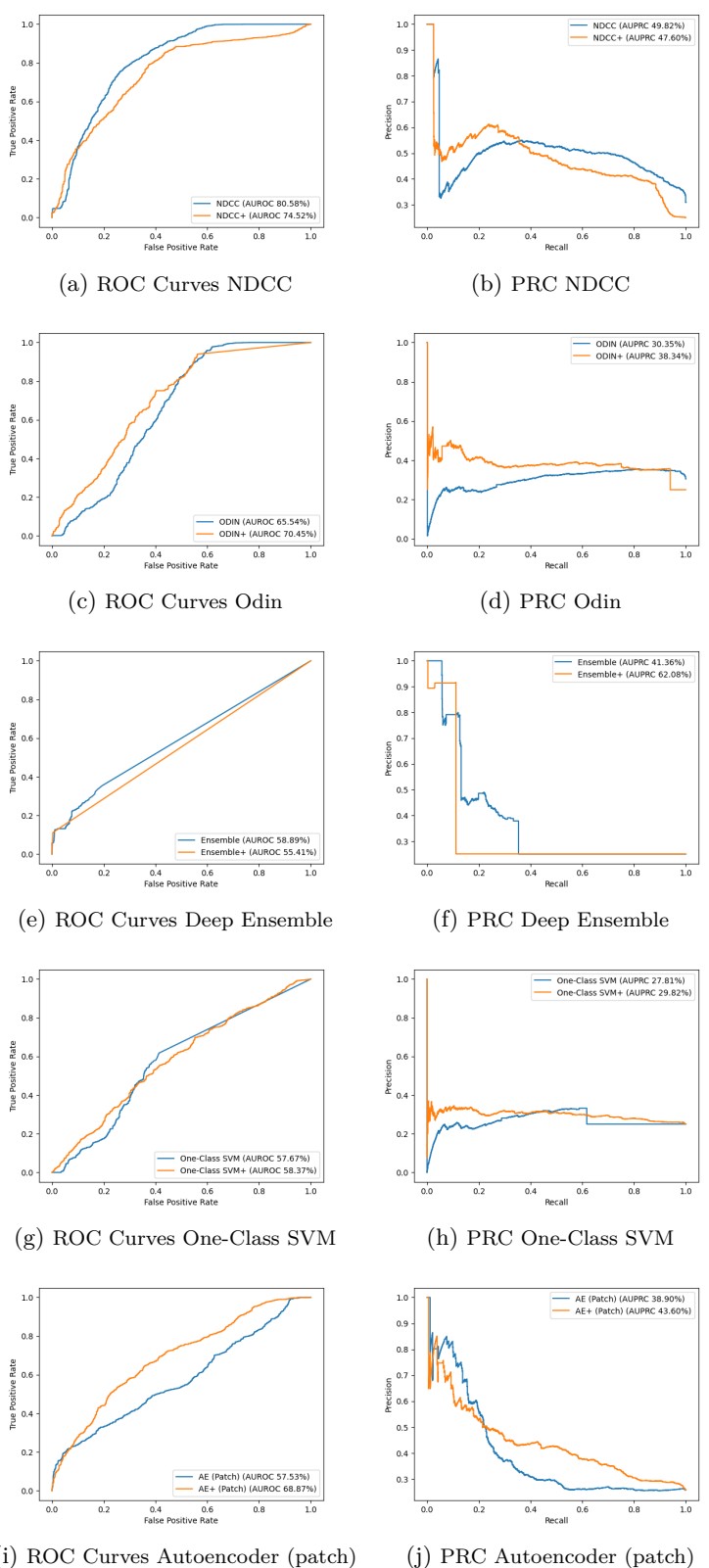

(a) ROC Curves NDCC

(b) PRC NDCC

(c) ROC Curves Odin

(d) PRC Odin

(e) ROC Curves Deep Ensemble

(f) PRC Deep Ensemble

(g) ROC Curves One-Class SVM

(h) PRC One-Class SVM

(i) ROC Curves Autoencoder (patch)

(j) PRC Autoencoder (patch)

Figure E.2: Novelty detection tradeoff curves (higher is better) on NovelCraft test data when trained with and without NovelCraft+.

