# OpenReview forum: "NovelCraft: A Dataset for Novelty Detection and Discovery in Open Worlds"
_TMLR — Accepted by TMLR_

### Review · Reviewer_bgxd · 2023-01-27

**Summary Of Contributions:**

The main contribution of this paper is the NovelCraft dataset, an open-world dataset composed of egocentric videos of an artificial agent building a pogo stick in a MineCraft-like environment. In some episodes, the world state has been modified such that novel types of objects  can be seen, and hence the primary goal of the dataset is to facilitate novelty detection. In addition to pixel-based video frames, the content of each frame can also accessed in the form of a json file, which acts as a second modality. Finally, the dataset can also be used for category discovery.


**Audience:**

No

**Broader Impact Concerns:**

Broader impact section is provided.

**Claims And Evidence:**

No

**Requested Changes:**

My comments all focus on adjustments that would strengthen the work substantially in my view.

Critical to securing my recommendation are my comments about the lack of comparison to other datasets in terms of baseline performance. From the provided benchmark data it is not clear to me whether this dataset is useful and interesting to the novelty detection community, which is critical when proposing a new benchmark. Less critical are my comments on the symbolic aspect of the proposed dataset.

In general, I think that if the main benchmark of novelty detection / category discovery from video frames should be the focus on this contribution, and I would encourage the authors to strengthen this part substantially.



**Strengths And Weaknesses:**

I have a number of comments, though in the interest of transparency I should point out to the authors that I am not an expert on novelty detection, and largely unfamiliar with the literature. Hopefully this information will help in addressing and understanding (the focus of) my concerns:

* The motivation for this type of this dataset is largely lacking. In several places it is claimed that "existing evaluations are too
object-focused", and yet it is not made clear why this is a problem? Similarly, compared to datasets in the related work on "Vision for open worlds", it is stated that "Compared to these works, our dataset is distinct in its multimodality and how images are gathered over time by a navigating agent.", yet it is not clear why this is an important distinction. The same holds true for related work on "Industrial datasets", where it is stated that "Unlike these works, our target scenario is an agent navigating open worlds over time, not a fixed camera on an assembly line" without further clarification. I suspect that the authors will want to argue that their dataset, being open-world, egocentric, and scene-centric, better resemble how humans perceive the world. And yet, the proposed datasets consists of artificial renderings of a video game, compared to real-world visual scenes, which is a step back. Is this not an important distinction also?

* Relatedly, no comparison is provided to other available datasets to highlight why NovelCraft is preferential, eg. in terms of the insights that can be derived. For example, Figure 2 reports novelty detection trade-off curves for several baselines on NovelCraft are shown. However, each of these baselines can also be evaluated on what are described as object-focused novelty detection tasks. Thus it is important to compare how these methods compare on these different datasets and point out differences. Perhaps the proposed NovelCraft is more difficult overall, or perhaps a different ordering among the considered baseline is observed when using NovelCraft? In other words, one should provide concrete evidence (or good arguments) why the community should start evaluating on NovelCraft as opposed to datasets that are currently available. I should note that the visual category discovery experiments provide a bit of evidence in this direction already, but it is missing for novelty detection from video frames, which is what I consider to be the "main track".

* What makes a dataset "open-world" and what not was quite unclear to me. Arguably, the agent that is used for recording the videos in NovelCraft is following reasonably fixed script, similarly to how video recordings of a self-driving car (eg. Waymo Open dataset) tend to follow a clear path. Thus aren't both instances of "open world" datasets in this case? How does "open-worldness" relate to the task of novelty detection, which is what this dataset is designed for?

* Relatedly, It is written that "We specifically selected this Minecraft-like world because it has several key attributes that are helpful toward our long-term goal.", after which long-range planning over time, interaction with other agents, and combinatorial action spaces are listed as such attributes. Why exactly are these attributes important for novelty-detection? What insights can be obtained because of NovelCraft having these properties compared to prior datasets?

* The multi-modal aspect of NovelCraft, i.e. the symbolic counter-part to video frames in the form of json files, appears somewhat adhoc to me. If the motivation of NovelCraft to more closely resemble how humans perceive and act in the visual world, there where does this form of data come in? A secondary modality in the form of written dialogue between agents or auditory information would have made more sense. Is it likely that progress on video-frames + json (even when converted to time-series) using NovelCraft will entail progress on multimodal novelty detection in human-like settings?

* "Examination of metrics informed by task-specific costs" is listed as a core contribution of this work, yet in practice this appears to amount to simply reporting results thresholded at 95% TPR; and at 95% PPV. Thus, I would not consider this a core contribution of the paper.

---

> ### Author Response · Authors · 2023-02-07
> **Review Response**
>
> We thank the reviewer for their helpful constructive comments. We’ve tried to respond to each below (and indicated where our paper itself has been updated). We look forward to further discussion to improve the paper.
>
>
> 1a RE motivation for this dataset
>
> There are numerous applications that will require an agent moving through a world, interacting with it, and processing visually complex scenes rather than object-focused images. For example, consider a robot designed for aid and rescue operations in disaster areas or an AI designed to help firefighters complete simulated training exercises in a virtual reality world. It is important to build benchmarks that directly assess the challenges of building such agents.
>
> In Table D.2 of the Appendix, we show that state of the art novelty detection methods for object-focused datasets, such as NDCC and ODIN, perform significantly worse on NovelCraft, a scene-focused dataset, than existing object-focused datasets. It is important to quantify the magnitude of this drop in performance, and also to realize that the relative ranking of methods may differ as we transition from object-focused to scene-focused and between metrics (area-under-curve metrics vs max TPR at a fixed PPV). In our view, the current novelty detection literature does not address these needs. We believe evaluations on scene-focused datasets that unfold over time are necessary to create novelty detection methods that are applicable to many of the realistic ways we’d like to build agents that can move and interact in an open world.
>
> 1b RE are we trying to resemble humans?
>
> We are not trying to present our dataset as more closely “resembl[ing] how humans perceive the world.” Instead, we aim for our dataset to enable the development of visual novelty detection methods for artificial agents that can move through the world. Most of the properties of previous datasets that we highlight in the related works section are meant to show what holds those datasets back from being used for this goal. We show in Table D.1 that many datasets have disconnected individual images. Only UBnormal has scenes changing over time. However, UBnormal utilizes a fixed camera and static background images instead of an egocentric view from an agent exploring the environment.
>
> 2 RE request for concrete comparison to other datasets
>
> We do find that NovelCraft is more difficult for novelty detection than object-focused datasets, as shown in Table D.2. We also find that state of the art models are substantially outperformed by simpler models on some metrics, as shown in Section 4.3 and Table 2. This latter point was not emphasized enough in our previous draft, and we have updated Section 4.3 and Table 2 accordingly.
>
> Our main argument for why NovelCraft should be used is more about a difference in kind than an increase in difficulty. Instead of focusing on a typical image classification setting, where images are object-focused and do not depict a scene evolving over time, we consider a setting where images are produced by an egocentric agent. NovelCraft enables experiments that emulate novelty detection occurring as an artificial agent is run, as seen in Section 5. The ability to contrast symbolic and visual novelty detection methods is also unique to our knowledge. We have updated the paper with a new Table 1 and additional text in Section 2 describing how our dataset compares with others.
>
> 3 RE what makes a dataset “open world”?
>
> We have updated (Sec 2) to clarify our definition of an “open world” dataset. We define open world datasets as a scene-focused dataset whose test set contains object classes or gameplay possibilities not seen in the training set. In an open world, the number of possible object classes is infinite, and as the agent explores more new things could always be encountered. Naturally, in any finite dataset (like ours) only finitely many new classes can be represented, but the key idea is to evaluate the agent’s ability to handle new stimuli never before seen in training. The Waymo Open dataset is scene-focused, with multiple objects in frame in a complex scene, but lacks a notion of introduction of novel classes in the test set.
>
> Traditional novelty detection often assumes novelty takes a specific form (e.g. a new class of the same kind/type). For example, evaluations of NDCC pursue fine-grained novelty detection, assessing NDCC’s ability to detect when a given image of a bird species is not like the species observed at training. In contrast, our “open world” view assumes much more is possible (you may see a new bird, or a new frog, or a new rival agent, or trees with distinct bark that yield different resources when chopped down).

---

> > ### Author Response · Authors · 2023-02-07
> > **Review Response 2**
> >
> > 4 RE how does our “long term goal” yield insights
> >
> > The “long-term goal” we are referring to there is “agents that fuse vision with other sensors to adapt to open worlds”. We emphasize that successful adaptation in the presence of novelty (rather than mere detection) is the long-term goal of the applications we are inspired by. We highlight properties like complex action spaces and agent interactions to show that our chosen POGO environment requires non-trivial adaptations by an agent and thus provides numerous challenges to reaching this goal. From the vision perspective, the complexity of the task ensures that there is a larger variety in images compared to something like an agent following a set path or simple instructions.
> >
> > One specific insight we hope our work imparts on the novelty detection community is that, for applications with agents interacting over time on a goal-oriented task, it is often more important to emphasize per-episode rather than per-frame evaluation of detection. In many applications, agents may not need to detect with 100% accuracy in every single frame, just often enough to identify the novelty from a few frames and determine how to respond to solve the task.
> >
> > 5 RE choice of multi-modal format
> >
> > There are a variety of other modalities we could choose, but the choice of a JSON that can be converted into a vector of numeric and categorical variables is very general. In terms of similarity to application settings, this could be analogous to settings where non-visual sensors are capturing information about the agent or other objects in the environment.
> >
> > 6 RE metrics contribution
> >
> > The majority of works in visual novelty detection only report AUROC, with few providing AUPRC and even fewer providing metrics at a set threshold. We believe that reporting so few metrics limits the ability to meaningfully compare models and that reporting only area under curve metrics limits understanding of how these models may perform when applied to real problems where a set threshold must be specified in advance. In Section 4.3, we show that relative rankings of methods can change depending on the metric of interest. Although the act of reporting performance across metrics appears simple, we list it as a contribution to highlight how our argument departs from the majority of the current literature and encourage future works to more thoroughly measure model performance.

---

> > ### Comment · Reviewer_bgxd · 2023-02-15
> > **Thank you for your comments.**
> >
> > * The way I interpret your motivation for this dataset is that this dataset is different along some dimensions compared to what is available in the literature and therefore a significant contribution. However, for several of these dimensions it is not clear to me how they affect the task of novelty detection. For example, you speak of “numerous applications that will require an agent moving through a world, interacting with it, and processing visually complex scenes rather than object-focused images”. My question then is: how is novelty detection from the perspective of an agent moving through a world and interacting with it different (and more difficult) from novelty detection from existing  datasets? I suppose I can see the distinction between scene and object focused in that novel objects in a scene focused dataset may be out of context or only appear to the side. But how does agent movement factor in, or agent interaction, or moving cameras? Why is novelty detection from disconnected images easier than from a continuous stream of images? As you point out there exist already scene focused datasets, and arguably their scenes are visually more complex than what Minecraft-like scenes can offer. I have similar reservations about the multi-modal aspect: why is this an important consideration for the task of novelty detection in particular?
> >
> > * Thank you for adding the comparison in Appending D.2. It indeed seems that NovelCraft is generally a bit harder than the “object-focused” datasets. I was quite surprised to see that Autoencoders may be preferred in applications requiring high precision on NovelCraft, while they clearly struggle even on “object-focused” datasets like CIFAR10 in Appendix D.2. Is the hypothesis that the AE approach mainly struggles with visual complexity, or are the scores in Appendix D.2. poorly calibrated, i.e. insufficiently taking into account high precision?
> >
> > * Going by your definition of Open World, it seems to me that Waymo Open is open world because in different parts of the city there is a solid chance at encountering entities that have never before been seen? In general I would expect the different kinds of entities in datasets like Waymo Open to have a very long tail, which as you point out is characteristic of Open World.

---

> > > ### Author Response · Authors · 2023-02-16
> > > **Response to Additional Comments**
> > >
> > > 1 RE Dataset Motivation with Respect to Novelty Detection
> > >
> > > Rather than focusing on how changes in our dataset affect the task of single image novelty detection or this task’s difficulty, we view our dataset as urging the community to consider novelty detection as encompassing a broader set of tasks. Multimodal novelty detection is the clearest example of this. The task of novelty detection utilizing data with two different modalities is broader than unimodal novelty detection and therefore necessitates new techniques. Broadening what metrics are reported for single image novelty detection, motivated by the consideration of task-specific costs, also reflects this goal.
> > >
> > > A long term goal of this paper is to work towards a new task where agents work in conjunction with novelty detection models as they perform actions. This type of task will often require data that has the qualities you mentioned (perspective of an agent moving through the world, agent interaction, moving camera, ect.), which are not present in previous datasets. We view these qualities as aspects of this new task rather than a way to increase the difficulty of single image novelty detection. We push towards this goal in Section 5 via our episodic evaluations and use of delay as an additional metric. Our results show that current novelty detection methods must be improved before this broader task becomes feasible.
> > >
> > > 2 RE Effectiveness of AE
> > >
> > > We do not believe that autoencoders struggle with visual complexity, especially with recent vision transformer autoencoder methods improving autoencoder performance significantly.
> > >
> > > Autoencoders tend to be outperformed by other models when evaluating novelty detection in terms of AUROC and AUPRC, which take averages over all thresholds. Only by examining high precision and high TPR regimes specifically do we find cases where autoencoders perform surprisingly well. The results in Appendix D.2 do not report performance under such regimes and thus insufficiently take into account a high precision regime as you suggest.
> > >
> > > 3 RE Waymo Open as Open World
> > >
> > > Waymo Open may contain entities in the test set that do not appear in the training set, but as far as we can tell from “Scalability in Perception for Autonomous Driving: Waymo Open Dataset” the dataset lacks the labels needed for it to be used as an open world dataset. For example, it is possible that a water fountain only appears in the test set. However this falls outside of the four annotated classes (vehicle, pedestrian, cyclist, sign) and therefore new labeling would be required to make use of the water fountain as a test set exclusive class. This differs from NovelCraft, which provides 44 test set exclusive classes to enable these open world evaluations.

---

### Review · Reviewer_q2q8 · 2023-01-31

**Summary Of Contributions:**

This paper presents NovelCraft: a benchmark for studying novelty detection and discovery. Unlike previous benchmarks, the images considered here are frames taken from episodes of a trained artificial agent solving a task in minecraft, and are consequently substantially different from images in e.g. cifar-10, that features an object at the center of the image. In addition, this benchmark has a multi-modal nature, where aside from visual information, they have symbolic information too, in the form of JSON strings containing information about the state of the world. Finally, its sequential nature is interesting, which is exploited by the symbolic models but not the vision models currently, allowing for interesting future work. The authors contribute a study of existing methods for novelty detection and discovery in different settings (they consider visual-only novelty detection, symbolic-only and multi-modal approaches, as well as the related problem of generalized category discovery) and in particular discuss the nuances specific to the metrics, most notably that simply looking at AUROC and AUPRC doesn’t surface interesting trade-offs between models at particular points that might matter most for some downstream applications (e.g. ensuring at least 95% precision).


**Audience:**

Yes

**Broader Impact Concerns:**

I don't have any concerns.

**Claims And Evidence:**

Yes

**Requested Changes:**

- Regarding the relationship to previous scene-focused datasets, it would be good to also discuss what are the limitations of NovelCraft compared to previous ones? In which scenario(s) (e.g problem setting, motivation or modeling approach) should someone pick a previous one over NovelCraft?

- Give a precise definition of ‘episode’

- In Section 3.2, the authors state the number of episodes collected for ‘standard episodes’, ‘inserted object modifications’ and ‘novelcraft+ extra standard episodes’ but not for ‘gameplay altering modifications’. Would be useful to know this information. Especially since this category has the highest variance in terms of the number of images per episode, perhaps more episodes are needed?

- Is there a benefit to ever using the term positive predictive value (PPV) instead of just using precision? If I understand correctly the two are exactly synonyms, so that would help readability.

- For the symbolic and multi-modal novelty detection task, why are only standard episodes available for validation, as opposed to ones with novelty? Section 3 says that some novel classes are held-out for validation - but why isn’t this the case in this specific task? Please clarify and if possible unify the methodology used throughout the paper.

- Similarly, it wasn’t clear to me why there is a discrepancy between visual novelty detection and its symbolic and multimodal counterpart in terms of one being frame-based and the other episode-based. Is there some fundamental reason that this has to be the case? I couldn’t tell from reading the paper. Again, please clarify and unify if possible.

- For the multimodal approach, I was surprised that the Autoencoder was chosen as the vision base detector. As the authors noted, it’s not the highest-ranked out of the vision baselines from the previous section. Is requiring more than one training class a no-go in this scenario? It wasn’t clear to me why. It also isn’t clear to me why e.g. NDCC in principle requires more than one training class? Couldn’t we estimate the single training class with a (single) Gaussian and use that Mahalanobis distance to get the novelty score?

- In Section 6.2, describe the GCD contrastive loss mentioned.

- For Section 6.2, consider evaluating additional models or baselines aside from GCD SSKM and DINO SSKM which is an ablation of the former, if I understood correctly

- It would also strengthen the paper to experiment with using pretrained models for the rest of the paper’s experiments, aside from just GCD

- Minor: unlabeled vs unlabelled - Both are fine but should be consistent. I’ve seen both versions in the paper.


Additional discussion points
=====================

- Is it perhaps too optimistic to give as input the full state of the world (in the symbolic information), including items that the agent wouldn’t observe? Why is this design decision made? Is it hard to narrow down only the aspects that are observable by the agent? This way the visual and symbolic information would be about the same ‘parts’ of the world. Maybe that can be a different ‘level of difficulty’. Though I do see from the empirical results that there is certainly room to improve and the task is by no means ‘solved’.

- Can the dataset be ‘biased’ depending on the agent chosen for navigation to generate the episodes/image frames (i.e. different agents would generate different datasets with different relative difficulty) - or is this not a concern when we are specifically interested in novelty detection problems?


**Strengths And Weaknesses:**

Strengths:
- The paper is very well-written and easy to follow (some small exceptions; see the next section)
- The proposed benchmark and related design choices are for the most part well-motivated (some exceptions; see the next section)
- Interesting analyses of metrics and how relative rankings of methods can differ based on application-specific priorities, and of the effect of class imbalance on performance for the category discovery task.

Weaknesses:
- Some methodological choices seem inconsistent and should be better motivated
- The GCD results include only one model (and an ablation of that model). Would be nice to have a more thorough empirical investigation there too.

---

> ### Author Response · Authors · 2023-02-07
> **Review Response**
>
> We thank the reviewer for their helpful constructive comments. We’ve tried to respond to each below (and indicated where our paper itself has been updated). We look forward to further discussion to improve the paper.
> Minor: unlabeled vs unlabelled - Both are fine but should be consistent. I’ve seen both versions in the paper.
> RE 1 dataset limitations
>
> Section 2 has been updated to suggest NovelCraft is not suited to methods not aimed at handling class imbalance, which challenges many methods, or collaboration with an agent, with many methods focused on more typical classification tasks.
>
> RE 2 episode definition
>
> Section 3.2 has been updated with a precise definition of episode.
>
> RE 3 number of episodes for “gameplay altering modifications”
>
> The number of episodes for “gameplay altering modifications” has been added. We do not think more episodes are needed as we have collected more episodes for these novelties than inserted item novelties and frames from these episodes more frequently depict novelties.
>
> RE 4 use of PPV
>
> We chose to use positive predictive value since abbreviating as PPV allowed us to better format our tables and match the format of other metrics (TPR, TNR, ect.). We also highlight that PPV is also known as precision when introducing the metric.
>
> RE 5 differing choice of class splits
>
> In Section 5 we have few novelties to work with due to inserted object novelties being trivial for a symbolic method. As we want no novelties shared between the validation and test sets, we chose to use a standard only validation set to maximize variety in the test set. Section 5.1 has been updated to communicate the reasoning for the different choices between benchmarks.
>
> RE 6 differing choice of time window
>
> The symbolic model recognizes changes in behavior in the agent and environment, which requires a consideration of time within an episode and rules out single frame evaluations. Additionally, the episode-based evaluations align with the evaluation of agents, where an agent is evaluated on its ability to solve a task in an episode instead of the individual decisions made each frame. Section 5 has been updated to better communicate the motivation for the differing methodologies.
>
> RE 7 use of autoencoder in Section 5
>
> The single class limitation arises from inserted object novelties being trivial for a symbolic method, limiting which classes can be used. As with the previous discussion of a standard-only validation set, a single class training set was chosen to maximize test set diversity.
>
> NDCC requires more than one training class because the model requires the use of cross-entropy loss. Without multiple classes for the cross-entropy loss, the model breaks down and would essentially only learn a function to map images as close as possible to an arbitrary point in latent space. The cross-entropy loss is also necessary for the model identifiability that enables the Mahalanobis distance in the latent space to be a meaningful measure of novelty.
>
> RE 8 GCD contrastive loss
>
> A description of the GCD contrastive loss has been added to Section 6.2.
>
> RE 9 additional GCD evaluations
>
> As the original GCD paper was only published in June 2022, competing GCD models only started being released in late November and December 2022. These newer GCD are incremental improvements on the original GCD paper. They use various methods to improve feature learning but still rely on semi-supervised K-means for test accuracy.
>
> We are currently looking into the possibility of updating the paper with an experiment using semi-supervised Gaussian mixture models, which have not been investigated in the current literature and would reflect our opinion that improvements in the prediction model are necessary to significantly improve upon the original GCD paper. We hope to complete these experiments by the end of this week.
>
> RE 10 model pretraining
>
> We do finetune pretrained models, but in terms of using a pretrained model without finetuning the only additional methods this could be applied to is OCSVM and autoencoder, as the other models require the use of a task-specific classification layer. OCSVM already does poorly, so we don’t think that results with a non-finetuned model would be of interest. In the case of an autoencoder, using a non-finetuned model would violate the assumption that the model has been trained on normal data necessary for novelty detection.
>
> RE 11 labelled vs labeled
>
> Thanks for catching this spelling inconsistency; it has been fixed.

---

> ### Author Response · Authors · 2023-02-07
> **Discussion Points Response**
>
> Additionally, in response to your discussion points:
>
> 1 RE on the full state of the world being given in symbolic information
>
> Is it perhaps too optimistic to give as input the full state of the world (in the symbolic information), including items that the agent wouldn’t observe? Why is this design decision made? Is it hard to narrow down only the aspects that are observable by the agent? This way the visual and symbolic information would be about the same ‘parts’ of the world. Maybe that can be a different ‘level of difficulty’. Though I do see from the empirical results that there is certainly room to improve and the task is by no means ‘solved’.
> We agree that the symbolic information is optimistic compared to what would be expected in more practical applications. We chose to provide the full state in our dataset due to it still providing a challenge, as seen in Section 5, and to provide future work the ability to choose what to give and not give the models. The full state also provides the ability for future work to provide more vision-related information through methods similar to our filtering scheme in Section 3.2, such as determining what is visually observed by the agent. We expect a reduction in the symbolic data would increase the difficulty of the task, although the symbolic information could also be used to provide additional visual information to make the task easier (such as semantic segmentation maps or hints about which objects are depicted).
>
> 2 RE on agent bias of the dataset
>
> We agree that the dataset is biased by the agent chosen to solve the task. A different choice of agent would change the distribution of the data, although the degree to which the distribution would shift depends on how significant the behavioral changes are. A change in difficulty is also possible based on agent choice. For example, an agent that chooses to look at the ground instead of straight ahead would provide much less useful visual data.
>
> We are not concerned by this bias in the context of our benchmarks since we are not claiming that a model trained on NovelCraft will generalize to other tasks or other agents. Instead, NovelCraft provides a benchmark on a specific agent and task. We have provided code that can be reused to collect data from another agent on this task or new tasks in a PolyCraft environment. Our data collection functionality uses a very simple API to communicate with an agent and the PolyCraft environment has its own interfaces for agents and task definitions.

---

> ### Author Response · Authors · 2023-02-14
> **Additional GCD Experiment**
>
> We have implemented and evaluated semi-supervised Gaussian mixture models (SSGMM) for GCD as proposed in our earlier comment. These experiments have been added to the most recent revision of the paper and show that SSGMM outperforms SSKM when using pretrained features and has comparable but slightly worse performance on GCD trained features.
>
> Please let us know if there are any other concerns we can address.

---

> > ### Comment · Reviewer_q2q8 · 2023-02-20
> > **thank you for the thorough responses**
> >
> > Thank you for the thorough responses and discussion. These clarifications and modifications strengthen the paper.
> >
> > RE: "Section 2 has been updated to suggest NovelCraft is not suited to methods not aimed at handling class imbalance, which challenges many methods, or collaboration with an agent, with many methods focused on more typical classification tasks." -- it would perhaps be helpful to state some applications where class balance isn't an issue and previous benchmarks may suffice, as opposed to applications where NovelCraft is more appropriate. Some nice examples are given to your responses to other reviewers, for example, which I think would be nice to add to the discussion of relationship to other benchmarks. Also, is class imbalance the only differentiating factor? In your response to reviewer bgxd, you mentioned that "There are numerous applications that will require an agent moving through a world" and gave an example, but I assume there are also applications that don't require this (can you given an example?) and for which a previous benchmark may be a better choice? Would be great to add that discussion to section 2.
> >
> > For the semi-supervised Gaussian mixture models, it would be good to clarify in the paper whether this model comes from the literature (or what exactly its connection is to models in the literature) as this wasn't clear to me from reading that paragraph.

---

> > > ### Author Response · Authors · 2023-02-21
> > > **Additional Revisions**
> > >
> > > Thank you for highlighting these issues with the revised text.
> > >
> > > We updated Section 2 to provide an example of when class balancing may be preferred and when static cameras are preferred. We also highlight UBNormal as a similar dataset utilizing static cameras.
> > >
> > > We have updated our description of SSGMM to clarify that it is a new extension of GMM with the semi-supervision methodology found in SSKM.

---

### Review · Reviewer_GBwG · 2023-02-01

**Summary Of Contributions:**

This paper proposes a new dataset, based on a Minecraft-like engine, to specifically assess the novelty detection abilities of vision models. They collect trajectories (images and groundtruth symbolic game state) of an agent solving a construction task. They then modify the environment to include game modifications (fences around trees or different tree types) and novel objects (52 total), and collect the result as their novelty detection dataset (splitting some of these modifications and novel objects into train/validation/test sets).
Finally using their data, they assess existing models on their novelty detection performance (using either visual information only or combinations of visual and symbolic) and generalized category discovery abilities.

Overall, the dataset covers an interesting niche for researchers interested in models’ abilities to handle novel objects in simulated environments, but it might be too limited in size and scope to reach its full potential (especially compared to MineDOJO and MineRL). I also feel that the model assessment part of the paper was slightly overemphasized, and I would have preferred the paper to spend more time discussing the dataset and their proposed benchmarks in more detail.

Given the “TMLR two questions” for submissions, I would answer yes to the 2nd one but will need more details to be assured on the 1st one.


**Audience:**

Yes

**Claims And Evidence:**

Yes

**Requested Changes:**

1. Provide all Train and Validation curves/final performance for the Visual Novelty Detection, Multimodal Novelty Detection and Generalized Category Discovery tasks.
2. Clarify why the dataset couldn’t be made larger.
3. Clarify the “Filtering to identify non-standard images” section in 3.2.
4. Clarify on the choice of “normal” classes and if this is expected to affect results if they were modified.
5. [if shared by other reviewers] rebalance the paper to cover the dataset in more details in the main text

**Strengths And Weaknesses:**

1. The paper targets a useful dataset niche, where there are non-trivial modifications performed in a dataset, but where there are still consistent behaviors and an expectation of transfer. Keeping it to a simulated virtual environment is a limitation but they address the drawback themselves early and effectively.
   1. However, I found it slightly too difficult to understand the scope and details of the modifications performed. In effect, Figure A.1 is necessary to understand what happened, as well as Table A.2, but these are both in the Appendix.
   2. There are not enough details about the “original” pogo stick construction task, and how complex it is. Obviously one would like to see videos, which hopefully a website can remedy, but the paper sometimes brushes over information as if they ought to be clear to the reader.
2. The number of modifications is small, and as it stands it is hard to know if one should expect them to be easily learnable.
   1. 2 gameplay modifications are performed, but there is no indication of their exact effect on behaviors with statistics or measurements.
   2. 52 objects are introduced, but again it is hard to tell if these should easily be discriminable, or if they share too many visual features to pose issues for humans to recognize (for example). There is no indication of their effect on the agent’s behavior.
   3. The dataset is small, judging by Table A.2. We’re talking around 50-100 frames for most objects. This is slightly confusing, given it should be trivial to run the data collection for longer and increase these numbers dramatically?
   4. There are no human baselines on the proposed benchmarks, which make it hard to assess their difficulty and validity.
3. The section on “Filtering to identify non-standard images” needs more unpacking.
   1. Why did you have to use Blender to assess object visibility?
   2. Don’t you have access to ground-truth information and the game engine state to basically assess this yourself? If not, then I am not convinced that PolyCraft was the best choice?
4. Sections 4, 5 and 6 cover quite a lot of ground on assessing existing models on several tasks built upon the dataset.
   1. The analysis performed is good, going beyond just single metric accuracy with the ROC & PR curves.
   2. Unfortunately, as mentioned above, I feel like the balance is a bit off, and it is unclear if the paper ought to be primarily about the dataset or this model assessment. I would personally have spent more time on the dataset (explaining decisions, showcasing behaviors, statistics, etc).
   3. Importantly, the results of most assessed methods are very poor on the Test sets (judging from Figure 2, also true for Table 3 and Figure 3 to an extent).
      1. This is actually quite interesting if true, given this demonstrates that the benchmark is hard and that current methods fail on it.
   4. However, there is no information about the performance on the Train and Validation sets, so right now this is confounded and it might just be that no model learns to classify the “classes” appropriately enough.
      1. If performance is also poor on train/valid, then this means the modifications aren’t discriminable enough and diminish the interest of the dataset quite drastically. Again, without human baselines it is hard to assess the usefulness of the dataset.
   5. There is no discussion of how the “normal” classes were selected, and why one should expect them to form a valid novelty detection task.
   6. I had to check in the Appendix to see that you were fine-tuning the VGG model on your tasks, I feel like this should be flagged in the main text.

---

> ### Author Response · Authors · 2023-02-07
> **Review Response**
>
> We thank the reviewer for their helpful constructive comments. We’ve tried to respond to each below (and indicated where our paper itself has been updated). We look forward to further discussion to improve the paper.
>
> 1 RE training and validation metrics
>
> We provide all of the metrics used on the training and validation sets in TensorBoard files currently provided in supplementary material. These record loss on these sets over time and accuracy for normal class classification for classifier models. As an example, we can see that the VGG classifiers get near perfect accuracy on the training set and ~96% accuracy on the validation set. If you would like these metrics included in the main paper, we can add summary tables in the main paper or appendix.
>
> 2 RE dataset size
>
> We should clarify that the addition of NovelCraft+ data significantly increases the size of the dataset from about 1.5 GB to 17 GB, as this currently is only mentioned in Table 1. As shown in Table 1, we chose to focus on data from the standard environment when increasing our dataset’s size. This is because collecting data for novel environments takes 4 times as much time as for the standard environment due to only ~25% of images depicting the novel object.
>
> The cost of data collection is higher for NovelCraft than for many other datasets based on virtual environments for two reasons. First, running and collecting images from a game engine is more computationally expensive than methods able to efficiently render single frames such as UBnormal. Second, running an agent that is attempting to solve a task by communicating with the game engine adds more computational complexity. So while we could in principle create new data in perpetuity, the amount we produced was what was feasible given our time and computational budgets combined with the challenges of using a complex game environment and task-solving agent. In terms of pure compute hours, the base NovelCraft data took approximately 2 weeks and the NoveCraft+ data took approximately a month.
>
> 3 RE clarifying image filtering
>
> While it may be possible to modify the game engine to render semantic segmentation maps, this would have been a complex task as game modding APIs do support overriding such low-level functionality. Writing custom rendering code for any complex game would be an engineering challenge even if we chose to work on a fully open source game.
>
> Therefore we chose to recreate the 3D environment in Blender given ground-truth information for object and agent positions from the symbolic representation. The downside of this approach was the necessity of searching for two unknown parameters, the camera height and field of view. As we used the semantic segmentation maps to filter images and not as labels for an object localization task, we feel a slight reduction in accuracy was worth significantly reducing the engineering effort required. We’d be happy to update Section 3.2 with this information if you think it would improve the paper.
>
> 4 RE choice of normal classes
>
> We have updated Section 3.2 to clarify our motivation for the choice of normal classes. The classes were chosen for their higher number of frames containing novelty to facilitate a reasonable number of images in the training, validation, and test sets. We would expect similar results if a different set of normal classes had at least 50 images per class, but otherwise there would likely be a decrease in performance due to a lack of training images.
>
> 5 RE more focus on the dataset
>
> Based on your feedback, we have updated the paper to provide more details on the pogostick task in Section 3.1 (comment 1.2) and provide more specifics about the environment modifications (comment 1.1). To address the latter point, we expanded our descriptions of the environment modifications in Section 3.2 and added references to a new Figure 1, similar to Figure A.1, to help visualize how the environment changes.
>
> Based on feedback from other reviewers, we have moved some of the content from Section D.1 to our new Table 1 and revision of Section 2 to better highlight our relation to other recent datasets.
>
> If there’s other specific changes you would like to rebalance the paper in favor of more coverage of the dataset, please let us know.
>
>
> In addition to these changes, we updated the paper to make it more clear that we’re fine-tuning the VGG model to address comment 4.6.

---

> > ### Comment · Reviewer_GBwG · 2023-02-14
> > **.**
> >
> > Dear Authors,
> >
> > Thanks a lot for the precisions, this makes a lot of sense.
> >
> > Wrt proposed changes:
> > - 1 RE: Yes please could you add these in the Appendix?
> > - 3 RE: Yes that'd be great to add this into the main text please (or appendix depending on length restrictions)
> > - 4 RE: That would be great to mention briefly in the main text / Appendix.

---

> > > ### Author Response · Authors · 2023-02-14
> > > **Comment Response**
> > >
> > > The requested changes have been added in our most recent revision. Please let us know if there are any other concerns we can address.

---

### Author Response · Authors · 2023-03-30
**Thanks to all reviewers!**

Thank you to the reviewers and the editors for your time and hard work in giving constructive criticism. We have submitted a camera-ready version with very minor edits, mostly cosmetic improvements with some added details to address some of the lingering questions a bit more completely (esp. about the JSON format and its potential to be ported to other situations).

We are happy with how the paper turned out thanks to your input. Much appreciated!

---

### Decision · Action_Editors · 2023-03-16

**Recommendation:** Accept as is

**Comment:**

The proposed Novelcraft dataset presents a new scene-based benchmark for novelty detection that differs fairly substantially from object-centric benchmarks. One interesting highlight is that the relative performance of state-of-the-art approaches changes in this domain, and simple auto-encoders end up being quite robust. The multi-modal aspect has potential, but I'm not sure whether a JSON file of the state is necessarily the way to go as it's unclear if a method built on this could be easily ported to other kinds of modalities.

Regardless, after providing new experiments and further clarifications, the reviewers mostly believe that the claims are sufficiently supported and that there would be interest from some members of the community. One reviewer was still ambivalent about the motivation - how the agent-centric view of this dataset specifically interacts with the novelty detection task. Some further discussion of this in the final draft would be helpful.

**Audience:**

Yes, researchers interested in novelty detection could be interested in this dataset if they are interested in using novelty detection methods in conjunction with an agent that acts within an environment.

**Claims And Evidence:**

Yes, for the most part, however as one reviewer points out there is still a lack of clarity on exactly how agency, video, etc. affect the novelty detection task. There do appear to be changes in ranking of different approaches, and surprisingly an auto-encoder does quite well on this dataset, so overall I think that the claims are sufficiently justified.